# Markovian Flow Matching: Accelerating MCMC with Continuous Normalizing Flows

**Alberto Cabezas**[*]
Department of Mathematics and Statistics
Lancaster University, UK
a.cabezasgonzalez@lancaster.ac.uk

**Louis Sharrock**[*]
Department of Mathematics and Statistics
Lancaster University, UK
l.sharrock@lancaster.ac.uk

**Christopher Nemeth**
Department of Mathematics and Statistics
Lancaster University, UK
c.nemeth@lancaster.ac.uk

## Abstract

Continuous normalizing flows (CNFs) learn the probability path between a reference distribution and a target distribution by modeling the vector field generating said path using neural networks. Recently, Lipman et al. [45] introduced a simple and inexpensive method for training CNFs in generative modeling, termed flow matching (FM). In this paper, we repurpose this method for probabilistic inference by incorporating Markovian sampling methods in evaluating the FM objective, and using the learned CNF to improve Monte Carlo sampling. Specifically, we propose an adaptive Markov chain Monte Carlo (MCMC) algorithm, which combines a local Markov transition kernel with a non-local, flow-informed transition kernel, defined using a CNF. This CNF is adapted on-the-fly using samples from the Markov chain, which are used to specify the probability path for the FM objective. Our method also includes an adaptive tempering mechanism that allows the discovery of multiple modes in the target distribution. Under mild assumptions, we establish convergence of our method to a local optimum of the FM objective. We then benchmark our approach on several synthetic and real-world examples, achieving similar performance to other state-of-the-art methods, but often at a significantly lower computational cost.

## 1 Introduction

The task of sampling from a probability distribution known only up to a normalization constant is a fundamental problem arising in a wide variety of fields, including statistical physics [51], Bayesian inference [25], and molecular dynamics [43]. In particular, let $\pi(\mathrm{d}x)$ be a target probability distribution on $\mathbb{R}^d$ with density $\pi(x)$ with respect to the Lebesgue measure of the form[1]

$$\pi(x) = \frac{\hat{\pi}(x)}{Z}, \tag{1}$$

where $\hat{\pi} : \mathbb{R}^d \to \mathbb{R}_+$ is a continuously differentiable function which can be evaluated pointwise, and $Z = \int_{\mathbb{R}^d} \hat{\pi}(x)\mathrm{d}x$ is an unknown normalizing constant. We are interested in generating samples from the target distribution $\pi$ in order to approximate integrals of the form $\pi[f] = \mathbb{E}_\pi[f(x)]$, where $f : \mathbb{R}^d \to \mathbb{R}$.

---

[*]Equal contribution

[1]In a slight abuse of notation, we use $\pi$ to denote both the target distribution and its density.

38th Conference on Neural Information Processing Systems (NeurIPS 2024).

A standard solution to this problem is Markov chain Monte Carlo (MCMC) [12, 64], which relies on the construction of a Markov process which admits the target $\pi$ as its invariant distribution. One of the most broadly applicable and widely studied MCMC methods is the Metropolis-Hastings (MH) algorithm [32], which proceeds in two steps. First, given a current sample $x$, a new sample $y$ is proposed according to some proposal distribution $q(\cdot|x)$. Then, this sample is accepted with probability $\alpha(x, y) = \min\left\{1, \frac{\pi(y)q(x|y)}{\pi(x)q(y|x)}\right\}$. This strategy generates a Markov chain with the desired stationary distribution and, under mild conditions on the proposal and the target, also ensures that the Markov chain is ergodic [65]. However, for high-dimensional, multi-modal settings, such methods can easily get stuck in local modes, and suffer from very slow mixing times [e.g., 48].

Naturally, the choice of proposal distribution $q(\cdot|x)$ is critical to ensuring that MH MCMC algorithms explore the target distribution within a reasonable number of iterations. A key goal is to obtain proposal distributions with fast mixing times, which can be applied generically to any target distribution. This is particularly challenging in the face of complex, multi-modal (or metastable) distributions, which commonly arise in applications such as genetics [38], protein folding [41], astrophysics [22], and sensor network localization [37]. On the one hand, local proposals, such as those employed in the Metropolis-Adjusted Langevin Algorithm (MALA) [66] or Hamiltonian Monte Carlo (HMC) [20, 56] struggle to transition between regions of high-probability, resulting in very long decorrelation times and few effective independent samples [e.g., 49]. On the other hand, global proposal distributions must be very carefully designed in order to avoid high rejection rates, particularly in high dimensions [17, 47].

Another popular approach to sampling is variational inference (VI) [10, 34, 61, 79], which obtains a parametric approximation $\pi_{\theta^*}(x) \approx \pi(x)$ to the target by minimising the Kullback-Leibler (KL) divergence to the target over a parameterized family of distributions $\mathcal{D}_\theta = \{\pi_\theta : \theta \in \Theta\}$. State-of-the-art VI methods use normalizing flows (NFs), which consist of a sequence of invertible transformations between a reference and a target distribution, to define a flexible variational family [62]. There has also been growing interest in the use of continuous normalizing flows (CNFs), which define a path between distributions using ordinary differential equations [15, 27, 45]. CNFs avoid the need for strong constraints on the flow but, until recently, have been hampered by expensive maximum likelihood training.

In recent years, several works have sought hybrid methods which utilize NFs to enhance the performance of MCMC algorithms; see, e.g., [28] for a recent review. For example, NFs have been successfully used to precondition complex Bayesian posteriors, significantly improving the performance of existing MCMC methods [e.g., 33, 39, 59, 68]. The synergy between local MCMC proposals and global, flow-informed proposals has also been explored, leading to enhanced mixing rates and effective estimation of multimodal targets [e.g., 24, 67].

**Our contributions**   In this paper, we continue this promising line of work, introducing a new probabilistic inference scheme which integrates CNFs with MCMC sampling techniques. Our approach utilizes flow matching (FM), a scalable, simulation-free training objective for CNFs recently introduced by Lipman et al. [45]. This enables, for the first time, the incorporation of CNFs into an adaptive MCMC algorithm. Concretely, our approach augments a local, gradient-based Markov transition kernel with a non-local, flow-informed transition kernel, defined using a CNF. This CNF, and the corresponding transition kernel, are adapted on-the-fly using samples from the chain, which are used to define the probability path for the FM objective. Our scheme also includes an adaptive tempering mechanism, which is essential for discovering multiple modes in complex target distributions. Under mild assumptions, we establish that the flow-network parameters output by our method converge to a local optimum of the FM objective. We then demonstrate empirically the performance of our approach on several synthetic and real-world examples, illustrating comparable or superior performance to other state-of-the-art sampling methods.

## 2   Preliminaries

**Continuous Normalizing Flows**   A continuous normalizing flow (CNF) is a continuous-time generative model which is trained to map samples from a base distribution $p_0$ to a given target distribution [15]. Let $v_t$ be a time-dependent vector field that runs continuously in the unit interval. Under mild conditions, this vector field can be used to construct a time-dependent diffeomorphic map

called a flow $\phi : [0, 1] \times \mathbb{R}^d \to \mathbb{R}^d$, defined via the ordinary differential equation (ODE):

$$\frac{\mathrm{d}}{\mathrm{d}t}\phi_t(x) = v_t(\phi_t(x)), \quad \phi_0(x) = x. \tag{2}$$

Given a reference density $p_0 : \mathbb{R}^d \to \mathbb{R}_+$, and the flow $\phi$, we can generate a probability density path $p : [0, 1] \times \mathbb{R}^d \to \mathbb{R}_+$ as the pushforward of $p_0$ under $\phi$, viz $p_t := [\phi_t]_\sharp p_0$, for $t \in [0, 1]$. This yields, via the instantaneous change-of-variables formula [e.g., 14]

$$\log p_t(x_t) = \log p_0(x) - \int_0^t \nabla \cdot v_s(x_s)\mathrm{d}s, \tag{3}$$

where $x_s := \phi_s(x)$, and where $\nabla$ is the divergence operator, i.e. the trace of the Jacobian matrix. In modern applications, the vector field $v_t$ is often parameterized using a neural network $v_t^\theta$, in which case the ODE in (2) is referred to as a neural ODE [15]. In turn, this yields a deep parametric model $\phi_t^\theta$ for the flow $\phi_t$, known as a CNF [27].

**Flow Matching**   One would typically like to learn a CNF which maps between a given reference density $p_0$ and a target density $\pi$. Given samples from the target, one approach is to maximize the log-likelihood $\mathbb{E}_{x \sim \pi}\left[\log p_1^\theta(x)\right]$. In practice, however, maximum likelihood training is very slow as both sampling and likelihood evaluation require multiple network passes to solve the ODE in (2).

Flow Matching (FM) provides an alternative, simulation-free method for training CNFs [45]. Let $p_t(x)$ be a target probability density path such that $p_0 = p$ is a simple reference distribution, and $p_1 \approx \pi$ is approximately equal to the target distribution. Let $v_t(x)$ be a vector field which generates this $p_t(x)$. Then the FM objective for the CNF vector field $v_t^\theta(x)$ is defined as

$$\mathcal{L}(\theta; \pi) = \mathbb{E}_{t \sim \mathcal{U}(0,1)}\mathbb{E}_{x \sim p_t}\left[\|v_t^\theta(x) - v_t(x)\|_2^2\right]. \tag{4}$$

In practice, we do not have direct access to the target vector field, $v_t(x)$, and so we cannot minimize (4) directly. However, as shown in Lipman et al. [45, Theorem 2], it is equivalent to minimize the conditional flow-matching (CFM) loss

$$\mathcal{J}(\theta; \pi) = \mathbb{E}_{t \sim \mathcal{U}(0,1)}\mathbb{E}_{x_1 \sim \pi}\mathbb{E}_{x \sim p_t(\cdot|x_1)}\left[\|v_t^\theta(x) - v_t(x|x_1)\|_2^2\right], \tag{5}$$

where $p_t(\cdot|x_1)$ is a conditional probability density path satisfying $p_0(x|x_1) = p_0$ and $p_1(x|x_1) \approx \delta_{x_1}$, and $v_t(\cdot|x_1) : \mathbb{R}^d \to \mathbb{R}^d$ is a conditional vector field that generates $p_t(\cdot|x_1)$. There are various choices for $p_t(\cdot|x_1)$ and $v_t(\cdot|x_1)$. For simplicity, we here assume that the conditional probability path is Gaussian, viz $p_t(x|x_1) = \mathcal{N}(x|m_t(x_1), s_t(x_1)^2 \mathbb{I}_d)$, where $m : [0, 1] \times \mathbb{R}^d \to \mathbb{R}^d$ denotes a time-dependent mean, and $s : [0, 1] \times \mathbb{R} \to \mathbb{R}_+$ a time-dependent scalar standard deviation. For our experiments, we further adopt the optimal transport conditional probability path introduced in [45], setting $m_t(x_1) = tx_1$ and $s_t(x_1) = 1 - (1 - \sigma_{\min})t$ for some $\sigma_{\min} \ll 1$. In this case, the conditional vector field assumes the particularly simple form $v_t(x|x_1) = \frac{x_1 - (1 - \sigma_{\min})x}{1 - (1 - \sigma_{\min})t}$.

## 3 Markovian Flow Matching

In this section, we present our main contribution, an adaptive MCMC algorithm which combines a non-local, flow-informed transition kernel trained via FM; a local, gradient-based Markov transition kernel; and an adaptive annealing schedule. We begin by describing how CNFs can be used within a MH MCMC algorithm.

### 3.1   MCMC with Flow Matching

Suppose, for now, that we have access to a CNF $(\phi_t^\theta)_{t \in [0,1]}$, trained (e.g.) via flow-matching, with corresponding vector field $(v_t^\theta)_{t \in [0,1]}$, which generates a probability path $(p_t^\theta)_{t \in [0,1]}$ between a reference density $p_0$ and an approximation of the target density $\pi$. Given a point $x_0 \in \mathbb{R}^d$ on the reference space, we can evaluate the log-density of the pullback of the target distribution $\pi$ as

$$\log[\phi_1^\theta]^\sharp \pi(x_0) = \log \pi(\phi_1^\theta(x_0)) - \int_1^0 \nabla \cdot v_t^\theta(\phi_t^\theta(x_0))\mathrm{d}t. \tag{6}$$

Under the assumption that the CNF approximately transports samples from $p_0$ to $\pi$, we expect that $[\phi_1^\theta]_\sharp p_0 \approx \pi$ in the target space, and that $[\phi_1^\theta]^\sharp \pi \approx p_0$ in the reference space. Given that the reference distribution $p_0$ is chosen such that it is easy to sample from, this suggests the following strategy, sometimes referred to as *neural trasport MCMC* or *neutraMCMC* [28, 33, 44, 59]. First, transform initial positions $x_1$ from the target space to the reference space by solving

$$\begin{bmatrix} x_0 \\ \log p_1^\theta(x_1) - \log p_0(x_0) \end{bmatrix} = \begin{bmatrix} x_1 \\ 0 \end{bmatrix} + \int_1^0 \begin{bmatrix} v_t^\theta(x_t) \\ -\nabla \cdot v_t^\theta(x_t) \end{bmatrix} \mathrm{d}t, \qquad (7)$$

which integrates the combined dynamics of $x_t$ and the log-density of the sample backwards in time. Then, generate MCMC proposals $y_0$ in the reference space using any standard MCMC scheme which targets the pullback of the target distribution, as defined in (6). Finally, transform accepted proposals back to target space using the forward dynamics, viz

$$\begin{bmatrix} y_1 \\ \log p_1^\theta(y_1) - \log p_0(y_0) \end{bmatrix} = \begin{bmatrix} y_0 \\ 0 \end{bmatrix} + \int_0^1 \begin{bmatrix} v_t^\theta(y_t) \\ -\nabla \cdot v_t^\theta(y_t) \end{bmatrix} \mathrm{d}t. \qquad (8)$$

This corresponds to using a transformation-informed proposal in a Markov transition step, an approach which has been successfully applied using (discrete) normalizing flows [28, 33, 44, 59].

There are various possible choices for the proposal distribution on the reference space (see Appendix A). For example, [23] consider an independent MH (IMH) proposal, where i.i.d. samples are drawn from the reference distribution. Here we focus on a flow-informed random-walk, motivated largely by its superior empirical performance in numerical experiments. This proposal performs particularly well on high-dimensional problems, where overfitting of the CNF can be corrected with stochastic steps, while exacerbated by independent proposals [39]. Concretely, our flow-informed random-walk transition kernel, summarized in Algorithm 2 (see Appendix A), can be written as

$$P(x, \mathrm{d}y; \pi, \theta) = \alpha(x, y)\rho_\theta(\mathrm{d}y|x) + (1 - b(x))\delta_x(\mathrm{d}y), \qquad (9)$$

where $\rho_\theta(\mathrm{d}y|x)$ is the distribution defined by the transition

$$x_0 = x + \int_1^0 v_t^\theta(\phi_t^\theta(x))\mathrm{d}t, \quad y_0 \sim \mathcal{N}(x_0, \sigma_{\mathrm{opt}}^2), \quad y = y_0 + \int_0^1 v_t^\theta(\phi_t^\theta(y_0))\mathrm{d}t, \qquad (10)$$

and $\alpha(x, y) = \min\left\{1, \frac{\pi(y)\rho_\theta(x|y)}{\pi(x)\rho_\theta(y|x)}\right\}$ and $b(x) = \int_{\mathbb{R}^d} \alpha(x, y)\rho_\theta(\mathrm{d}y|x)$.

**Training the CNF** Thus far, we have assumed that it is possible to train a CNF which maps samples from the reference distribution $p_0$ to (an approximation of) the target distribution $\pi$. Clearly, however, the CFM objective is not immediately applicable in our setting, since we do not have access to samples from the target $\pi$.

Tong et al. [72] propose two alternatives in this case: (i) use an importance sampling reweighted objective function, or (ii) use samples from a long-run MCMC algorithm (e.g., MALA) as approximate target samples. Both of these approaches, however, have limitations. The former is unlikely to succeed when the proposal distribution differs significantly from the target distribution, while the latter will only perform well when the chosen MCMC method mixes well.

In this paper, we adopt a different approach, updating the parameters of a CNF based on a dynamic estimate of the CFM objective obtained via an adaptive MCMC algorithm. This is similar in spirit to other recent flow-informed MCMC algorithms [24, 35, 67], and the *Markovian score climbing* algorithm in [54].

## 3.2 Adaptive MCMC with Flow Matching

**Overview** Our adaptive MCMC scheme combines a non-local, flow-informed transition kernel (e.g., a flow informed random-walk) and a local transition kernel (e.g., MALA), which generate new samples from a sequence of annealed target distributions. These new samples are used to define a new estimate of the CFM objective in (5), which is optimized to define a new CNF. These steps are repeated until the samples converge in distribution to the target $\pi$, and the flow-network parameters converge to a local minima of the flow matching objective (see Proposition 3.1). This scheme, which we refer to as *Markovian Flow Matching* (MFM), is summarized in Algorithm 1.

**Sampling** There is significant freedom regarding the choice of both the local and the non-local MCMC algorithms. In our experiments, we adopt the Metropolis-Adjusted Langevin Algorithm (MALA) as the local algorithm. Thus, the local Markov kernel $Q$ is given by

$$Q(x, \mathrm{d}y; \pi) = \alpha(x, y)q(\mathrm{d}y|x) + (1 - b(x))\delta_x(\mathrm{d}y), \qquad (11)$$

where $q(\mathrm{d}y|x)$ is given by

$$q(\mathrm{d}y|x) \propto \exp\left(-\frac{1}{4\tau}\|y - x - \tau\nabla\log\pi(x)\|^2\right)\mathrm{d}y, \qquad (12)$$

and where, as elsewhere, $\alpha(x, y) = \min\left\{1, \frac{\pi(y)q(x|y)}{\pi(x)q(y|x)}\right\}$ and $b(x) = \int_{\mathbb{R}^d}\alpha(x, y)q(\mathrm{d}y|x)$. In principle, however, other choices such as HMC could also be used.

Meanwhile, for the non-local MCMC algorithm, we adopt the flow-informed random walk with non-local Markov kernel $P$ defined in (9). Together, assuming alternate local and non-local steps, these two kernels define a Markov chain with Markov transition kernel $R := P \circ Q$, given explicitly by $R(x, \mathrm{d}y; \pi, \theta) = \int_{\mathcal{Z}} Q(x, \mathrm{d}z; \pi)P(z, \mathrm{d}y; \pi, \theta)$. In practice, the balance between local and non-local moves is controlled by the hyperparameter $k_Q$, which sets the number of local steps before a global step.

**Training** Following each MCMC step, the parameters of the flow-informed Markov transition kernel $P$ are updated based on a new estimate of $\mathcal{J}(\theta; \pi)$. To be precise, suppose we write $\mu_t := \mu_0 R^k(\cdot, \cdot; \pi, \theta)$ for the distribution of the Markov chain with kernel $R(\cdot, \cdot; \pi, \theta)$ after $k \in \mathbb{N}$ steps, starting from initialization $\mu_0$, where $R^k = R \circ R \cdots \circ R$. Our objective function is then given by

$$\mathcal{J}(\theta; \mu_k) = \mathbb{E}_{t\sim\mathcal{U}(0,1)}\mathbb{E}_{x_1\sim\mu_k}\mathbb{E}_{x\sim p_t(\cdot|x_1)}\left[\|v_t^\theta(x) - v_t(x|x_1)\|_2^2\right]. \qquad (13)$$

For our choice of conditional probability path (i.e., the optimal transport path), we can in fact rewrite this objective as [45, Section 4.1]

$$\mathcal{J}(\theta; \mu_k, \sigma_{\min}) = \mathbb{E}_{t\sim\mathcal{U}(0,1)}\mathbb{E}_{x_1\sim\mu_k}\mathbb{E}_{x_0\sim p_0}\left[\|v_t^\theta(\phi_t(x_0|x_1)) - v_t(\phi_t(x_0|x_1)|x_1)\|_2^2\right], \qquad (14)$$

where $v_t(x|x_1) = \frac{x_1 - (1-\sigma_{\min})x}{1-(1-\sigma_{\min})t}$ and $\phi_t(x|x_1) = (1 - (1-\sigma_{\min}t)x + tx_1$. In practice, we will optimize a Monte Carlo estimate of this objective, namely,

$$\mathcal{J}(\theta; \{x^i(k)\}_{i=1}^N, \sigma_{\min}) = \frac{1}{N}\sum_{i=1}^N\|v_{t_i}^\theta(\phi_{t_i}(x_0^i|x^i(k))) - v_{t_i}(\phi_{t_i}(x_0^i|x^i(k))|x^i(k))\|_2^2, \qquad (15)$$

where $\{x^i(k)\}_{i=1}^N$ are the samples from $N$ chains of our MCMC algorithm after $k \in \mathbb{N}$ iterations, $x_0^i \overset{\text{i.i.d.}}{\sim} p_0$, and $t_i \sim \mathcal{U}(0, 1)$. The use of $N$ particles allow the state's mutation to $N$ computing cores running in parallel at each iteration. Sampling steps can be run in parallel using modern vector-oriented libraries, before each particle is used to approximate the loss and update the parameters. Thus, the speedup gained by using more than one core scales linearly with the number of cores as long as there are as many cores as there are particles.

**Annealing** For complex (e.g., multimodal) target distributions, it can be challenging to learn a CNF that successfully maps between the reference $p_0$ and the target $\pi$. For example, if the locations of the modes of the target are not known a priori, and the MCMC chains are initialized far from one or more of the modes, it is unlikely that the local MCMC kernel, and therefore the trained flow, will ever discover these modes [e.g., 24, Section IV.C]. To alleviate this problem, one approach is to iteratively target a sequence of annealed densities $\{\pi_k(x)\}_{k=0:K}$, which smoothly interpolate between a simple base distribution $\pi_0(x)$ (e.g., a standard Gaussian), and the target distribution $\pi_K(x) := \pi(x)$. This idea is central to other Monte Carlo sampling methods such as Sequential Monte Carlo (SMC) [18] and Annealed Importance Sampling (AIS) [55], as well as sampling methods used in score-based generative modelling [e.g., 70]. In our case, the annealed targets act as intermediary steps within the flow-informed MCMC scheme.

A standard way in which to construct the sequence $\{\pi_k(x)\}_{k=0:K}$ is to use a geometric interpolation, defining

$$\pi_k(x) = \pi_K(x)^{\beta_k}\pi_0(x)^{1-\beta_k}, \qquad (16)$$

---

**Algorithm 1** Markovian Flow Matching

---

1: **Input:** target $\pi$, base $\pi_0$, vector field $v_t^\theta$, reference $p_0$, concentration $\sigma_{min}$, initial parameters $\theta_0$, target ESS fraction $\alpha$, iterations $K$, number of particles $N$, local kernel $Q$, flow-informed kernel $P$, MCMC steps per resampling step $k_Q$, step sizes $\varepsilon_{1:K}$.
2: **Output:** flow-network parameters $\theta_K$
3: Sample $x_0^i \sim \pi_0$ for $i = 1, \ldots, N$ (*initialize samples*)
4: **for** $k = 1 : K$ **do**
5:     **if** $\beta_{k-1} < 1$ **then**
6:         $\beta_k =$ solve (17) for $\beta_{k-1} < \beta \leq 1$ (*update annealing temperature*)
7:         $\pi_k =$ solve (16) (*update annealing density*)
8:     **end if**
9:     **if** $k \mod k_Q + 1 = 0$ **then**
10:         $x_k^i \sim P(x_{k-1}^i, \cdot; \pi_k, \theta_{k-1})$ for $i = 1, \ldots, N$ (*flow-informed Markov transition*).
11:     **else**
12:         $x_k^i \sim Q(x_{k-1}^i, \cdot; \pi_k)$ for $i = 1, \ldots, N$ (*local Markov transition*).
13:     **end if**
14:     $\theta_k = \theta_{k-1} + \varepsilon_k \nabla_\theta \mathcal{J}(\theta_{k-1} | \{x_k^i\}_{i=1}^N, \sigma_{\min})$ (*update flow-network parameters*)
15: **end for**

---

where $\beta_{0:K}$ is a sequence of temperatures which satisfies $0 = \beta_0 < \beta_1 < \cdots < \beta_K = 1$ [e.g., 55]. In practice, it can be difficult to choose a good sequence of temperatures that provides a smooth transition between densities. One heuristic for adaptively setting this sequence is based on the effective sample size (ESS). In particular, by setting the ESS to a user-specified percentage $\alpha$ of the number of particles $N$, the next temperature $\beta_k$ in the schedule can be determined by solving the recursive equation [9]

$$\beta_k = \inf \left\{ \beta_{k-1} < \beta \leq 1 : \frac{\left[ \frac{1}{N} \sum_{i=1}^N w_i^{\beta_{k-1}}(\beta) \right]^2}{\frac{1}{N} \sum_{i=1}^N w_i^{\beta_{k-1}}(\beta)^2} = \alpha \right\}, \qquad (17)$$

where $w_i^{\beta_{k-1}}(\beta) = \left[ \pi_K(x^i)^\beta \pi_0(x^i)^{1-\beta} \right] / \left[ \pi_K(x^i)^{\beta_{k-1}} \pi_0(x^i)^{1-\beta_{k-1}} \right] = \left[ \pi_K(x^i)/\pi_0(x^i) \right]^{\beta - \beta_{k-1}}$ are new importance weights given the current temperature $\beta_{k-1}$. In practice, we find that the inclusion of this adaptive tempering scheme is essential in the presence of highly multimodal target distributions, enabling the discovery of modes which are not known *a priori*.

**Convergence** The output of Algorithm 1 is a vector of parameters $\theta_K$ which defines a CNF $(\phi_t^{\theta_K})_{t \in [0,1]}$. Under the assumption that the parameter estimate converges, that is, $\theta_K \to \theta_{\text{global}}^*$ as $K \to \infty$, where $\theta_{\text{global}}^* = \arg \min_{\theta \in \Theta} \mathcal{J}(\theta; \pi)$ is the global minimizer of the CFM objective $\mathcal{J}(\theta; \pi)$ in (5), this CNF is guaranteed to generate a probability path $(p_t^\theta)_{t \in [0,1]}$ which transports samples from the reference $p_0$ to the true target $\pi$ [e.g., 45].

In practice, the objective $\mathcal{J}(\theta; \pi)$ is highly non-convex, and thus it is not possible to establish a convergence result of this type without imposing unreasonably strong assumptions on the vector field $(v_t^\theta)_{t \in [0,1]}$. This being said, it is reasonable to ask whether $\theta_K$ converges to a local optimum of the CFM objective. We now answer this question in the affirmative. In particular, under mild regularity conditions, Proposition 3.1 guarantees that $\theta_K \to \theta^*$ almost surely as $K \to \infty$, where $\theta^*$ denotes a local minimum of the CFM objective. This proposition closely mirrors [54, Proposition 1]. Its proof, which relies on a classical result in [6, Theorem 3.17], is provided in Appendix B.

**Proposition 3.1.** *Assume that Assumptions B.1 - B.6 hold (see Appendix B). Assume also that $(\theta_K)_{K \in \mathbb{N}}$ is a bounded sequence, which almost surely visits a compact subset of the domain of attraction of $\theta^*$ infinitely often. Then $\theta_K \to \theta^*$ almost surely.*

## 4 Related work

In recent years, a number of works have proposed algorithms which combine MCMC techniques with NFs; see, e.g., [2, 28] for recent surveys. Broadly speaking, these algorithms fall into two distinct categories. *NeutraMCMC* methods leverage NFs as reparameterization maps which simplify

the geometry of the target distribution, before running (local) MCMC samplers in the latent space. This technique was pioneered in the seminal paper [59], and since been investigated in a number of different works [e.g., 13, 33, 44, 54, 58, 68, 82, 84]. *Flow MCMC* methods, meanwhile, utilize the pushforward of the base distribution through the NF as an (independent) proposal within an MCMC scheme. This approach was first studied by [3], and further extended in [4, 31, 57].

More recently, [24, 67] have introduced adaptive MCMC schemes which combine local MCMC samplers (e.g., MALA or HMC), with a non-local, flow-informed proposal (IMH or i-SIR); see also [35]. Our algorithm combines aspects of both *neutraMCMC* and *flow MCMC* methods and, unlike any existing approach, make use of a CNF (as opposed to a discrete NF), by leveraging the conditional flow matching objective. The use of NFs within other Monte Carlo algorithms has also been the subject of recent interest. For example, [5, 50] consider augmenting SMC with NFs, while [19, 52] use NFs (or diffusion models) within AIS.

Although less directly comparable to our own approach, several other recent works have proposed to use (controlled) diffusion processes to sample from unnormalized probability distributions. Such works include Zhang and Chen [85], who introduce the *path integral sampler*, Vargas et al. [75], who propose the *denoising diffusion sampler*, and Zhang et al. [83], who introduce *generative flow samplers*. Some other relevant contributions in this direction include [1, 8, 16, 60, 63, 69, 73, 74, 75, 76, 77, 78].

# 5 Experiments

In this section, we evaluate the performance of MFM (Algorithm 1) on two synthetic and two real data examples. Our method is benchmarked against four relevant methods. The Denoising Diffusion Sampler [DDS; 75] is a VI method which approximates the reversed diffusion process from a reference distribution to an extended target distribution by minimizing the KL divergence. Adaptive Monte Carlo with Normalizing Flows [NF-MCMC; 24] is an augmented MCMC scheme which uses a mixture of MALA and adaptive transition kernels learned using discrete NFs. Flow Annealed Importance Sampling Bootstrap [FAB; 52] is an augmented AIS scheme minimizing the mass-covering $\alpha$-divergence with $\alpha = 2$. Finally, Adaptive Tempered SMC (AT-SMC), i.e. the SMC algorithm described in [18] using a MALA transition kernel and a sequence of annealed distributions chosen adaptively by solving (17).

For each experiment, all MALA kernels use the same step size, targeting an acceptance rate of close to 1 since we estimate expectations, e.g. in (14), using the current ensemble of particles, rather than a single long chain. Following [85], we parameterize the vector field as

$$\text{NN}^*(t; \theta_3) v_t^\theta(x) = \text{NN}(x, t; \theta_1) + \text{NN}(t; \theta_2) \times \nabla \log \pi(x), \tag{18}$$

where the neural networks are standard MLPs with 2 hidden layers, using a Fourier feature augmentation for $t$ [71], and where $\text{NN}^*$ outputs a real value that reweights the vector field output using the time component. This architecture is also used by DDS [75, Section 4]. Meanwhile, FAB and NF-MCMC use rational quadratic splines [21]. Flows are trained using Adam [40] with a linear decay schedule terminating at $\varepsilon_K = 0$. We report results for all methods averaged over 10 independent runs with varying random seeds. Code to reproduce the experiments is provided at https://github.com/albcab/mfm.

## 5.1 4-mode Gaussian mixture

Our first example is a mixture of four Gaussians, evenly spaced and equally weighted, in two-dimensional space. The four mixture components have means $(8, 8)$, $(-8, 8)$, $(8, -8)$, $(-8, -8)$, and all have identity covariance. This ensures that the modes are sufficiently separated to mean that jumping between modes requires trajectories over sets with close to null probability. Given the synthetic nature of the problem, we can measure approximation quality using the Maximum Mean Discrepancy (MMD) [e.g., 29]; see Appendix C.1 for details. We can also include, as a benchmark, the results for an approximation learned using FM with true target samples. Diagnostics for all models are presented in Table 1, and learned flow samples in Figure 1. Further algorithmic details and results are provided in Appendix C.2.

In this experiment, only our method (Figure 1a) and DDS (Figure 1c) learn the fully separated modes, reflecting the greater expressivity of CNFs in comparison to the discrete NFs used in, e.g., NF-MCMC

| | 4-mode | | 16-mode | |
|---|---|---|---|---|
| | MMD | seconds | MMD | seconds |
| FM w/ $\pi$ samples | 3.69e-4±1.84e-4 | 22.3 ± 0.64 | 1.35e-3±6.66e-4 | 22.4 ± 1.01 |
| MFM $k_Q = K$ | 2.37e-3±2.29e-3 | 27.9 ± 1.27 | 1.88e-2±3.67e-3 | 28.2 ± 2.84 |
| MFM $k_Q = 10$ | 8.13e-4±4.41e-4 | 117. ± 5.65 | 2.87e-3±9.67e-4 | 89.6 ± 5.19 |
| DDS | 1.76e-4±2.32e-4 | 114. ± 0.68 | 1.02e-1±4.10e-2 | 115. ± 0.64 |
| NF-MCMC | 5.85e-3±3.91e-3 | 72.0 ± 11.7 | 8.05e-3±1.42e-2 | 67.0 ± 12.3 |
| FAB | 2.69e-4±2.06e-4 | 101. ± 3.24 | 1.51e-3±1.06e-3 | 102. ± 4.32 |
| AT-SMC | 3.95e-2±2.90e-2 | 2.18 ± 0.26 | 1.73e-2±5.30e-3 | 2.19 ± 0.21 |

Table 1: Diagnostics for the two synthetic examples. MMD is the Maximum Mean Discrepancy between real samples from the target and samples generated from the learned flow. Results are averaged and empirical 95% confidence intervals over 10 independent runs.

(Figure 1d). It is worth noting that DDS provides a closer approximation to the real target than MFM and, notably, even FM trained using true target samples (top row). Given that both methods use the same network architecture but a different learning objective, this suggests a potential limitation with the FM objective, at least when using this network architecture. This being said, MFM is notably more efficient than DDS (as well as the other methods) in terms of total computation time. While this is not a critical consideration in this synthetic, low-dimensional setting, it is a significant advantage of MFM in higher-dimensional settings involving real data (e.g., Section 5.3 and Section 5.4).

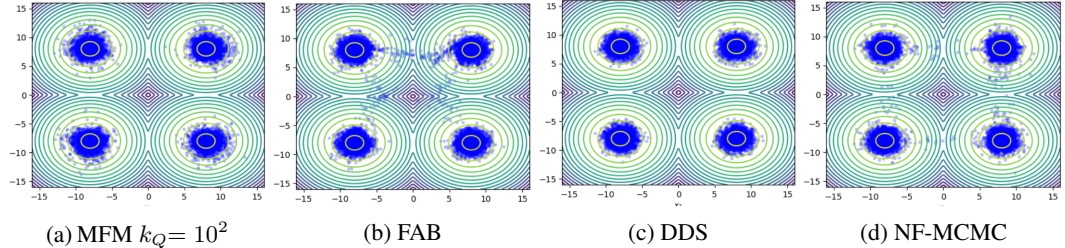

(a) MFM $k_Q = 10^2$      (b) FAB      (c) DDS      (d) NF-MCMC

Figure 1: Comparison between MFM, FAB, DDS, and NF-MCMC. Samples from the target density for the 4-mode Gaussian mixture example.

## 5.2 16-mode Gaussian mixture

The second experiment is a mixture of bivariate Gaussians with 16 mixture components. This is a modification of the 4-mode example, with contrasting qualities that illustrate other characteristics of each of the presented methods. In this case, the modes are evenly distributed on $[-16, 16]^2$, with random log-normal variances. The number of modes reduces the size of sets of (near) null probability between the modes, making jumping between them easier. To increase the difficulty of this model, all methods are initialized on a concentrated region of the sampling space. Diagnostics are presented in Table 1 and learned flow samples in Figure 2. Further details are provided in Appendix C.3.

In this example, DDS collapses to the modes closest to the initial positions while our method captures the whole target. Since the modes are no longer separated by areas of near-zero probability, the discrete NF methods are now able to accurately capture the target density. In this case, FAB marginally outperforms MFM as measured by the MMD, but this slight improvement in performance comes at the cost of a much higher run-time.

## 5.3 Field system

Our first real-world example considers the stochastic Allen–Cahn model [7], used as a benchmark in [24], and described in Appendix C.5. This fundamental reaction-diffusion equation is central to the study of phase transitions in condensed matter systems. Incorporating random forcing terms or thermal fluctuations allows for a stochastic treatment of the dynamics, capturing the inherent randomness and uncertainties in physical systems. This model leads to a discretized target density

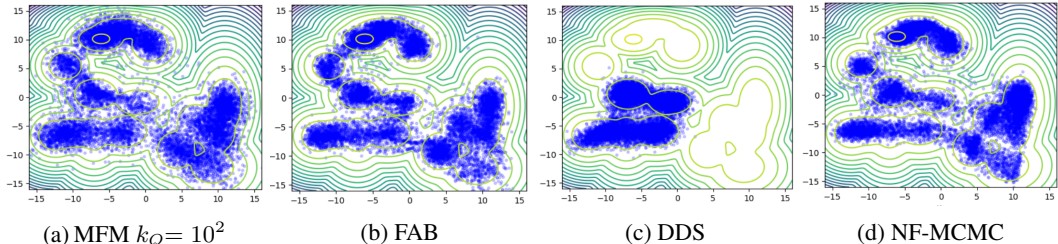

(a) MFM $k_Q = 10^2$       (b) FAB       (c) DDS       (d) NF-MCMC

Figure 2: Comparison between MFM, FAB, DDS, and NF-MCMC. Samples from the target density for the 16-mode Gaussian mixture example.

which takes the form

$$\log \pi(x) = -\beta \left( \frac{a}{2\Delta s} \sum_{i=1}^{d+1} (x_i - x_{i-1})^2 + \frac{b\Delta s}{4} \sum_{i=1}^{d} (1 - x_i^2)^2 \right), \tag{19}$$

with $\Delta s = \frac{1}{d}$, and boundary conditions $x_0 = x_{d+1} = 0$. In our experiments, we take $d = 64$. Meanwhile, following [24], other parameter values are chosen to ensure bimodality at $x = \pm 1$: $a = 0.1$, $b = 1/a = 10$, and $\beta = 20$. The bimodality induced by the two global minima complicates mixing when using traditional MCMC updates. Learning the global geometry of the target and using that information to propose transitions facilitates movement between modes. Unlike previous work [e.g., 24], we deliberately choose not to employ an informed base measure. Instead, we opt for a standard Gaussian with no additional information, making the problem significantly more challenging. This choice illustrates the robustness of our approach.

Numerical diagnostics for each method are presented in Table 2. In this case, we use the Kernelized Stein Discrepancy (KSD) as a measure of sample quality [e.g., 26, 46]; see Appendix C.1 for details. While this is not a perfect metric, it does allow us to qualitatively compare the different methods considered.

In this case, the tempering mechanism of our method is crucial for ensuring that the learned flow does not collapse on one of the modes and instead explores both global minima. This is confirmed when plotting the samples generated in the grid in Figure 3. This experiment demonstrates the ability of our method to capture complex multi-modal densities, even without an informed base measure, at a *significantly* lower computational cost (e.g., 10-25x faster) than competing methods. Indeed, while FAB was the best performing method in this experiment as measured by the KSD, it failed to capture both of the modes in the target distribution, and required a much greater total computation time (see Table 2).

It is worth noting that MFM (and the other two benchmarks, DDS and FAB) significantly outperformed NF-MCMC in this example, despite the similarities between MFM and NF-MCMC. While we tested various hyperparameter configurations for NF-MCMC, we were not able to find a setting that achieved comparable results in the absence of an informed base measure.

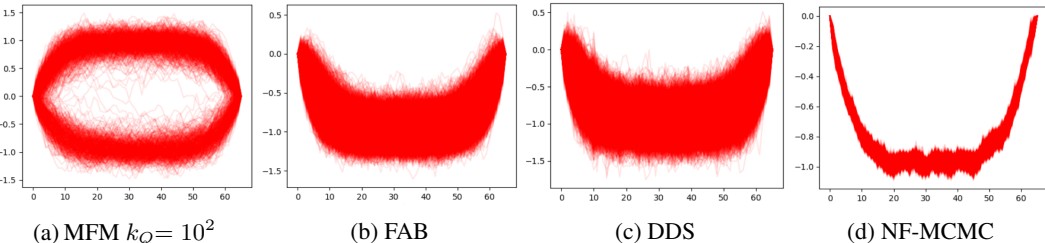

(a) MFM $k_Q = 10^2$       (b) FAB       (c) DDS       (d) NF-MCMC

Figure 3: Comparison between MFM, FAB, DDS, and NF-MCMC. Representative samples from the target density for the Field system example.

## 5.4 Log-Gaussian Cox process

Bayesian inference for high-dimensional spatial models is known to be challenging. One such model is the log-Gaussian Cox process (LGCP) introduced in [53], which is used to model the locations of 126 Scots pine saplings in a natural forest in Finland. See Appendix C.6 for full details. The target space is discretized to a $M = 40 \times 40$ regular grid, rendering the target dimension $d = 1600$. In Table 2, we report diagnostics for each algorithm.

In this case, the lack of multimodality in the target makes it a good fit for non-tempered schemes. Similar to the previous example, NF-MCMC is unable to obtain an accurate approximation to the target distribution. We suspect that this may be a result of non-convergence: due to memory issues, it was not possible to run NF-MCMC (or FAB) for more than $K = 10^3$ iterations. This also explains the (relatively) smaller run times of these algorithms in this example. By a small margin, DDS provides the best approximation of the target, slightly outperforming MFM and FAB. Meanwhile, MFM provides a good approximation to the target at a lower computational cost with respect to its competitors.

|  | Field system | | | Log-Gaussian Cox | | |
|---|---|---|---|---|---|---|
| $K = 10^4$ | KSD U-stat. | KSD V-stat. | seconds | KSD U-stat. | KSD V-stat. | seconds |
| MFM $k_Q = K$ | $2.61 \pm 2.00$ | $20.9 \pm 2.49$ | $52.3 \pm 1.23$ | 1.13e-1$\pm$.05 | $28.1 \pm 0.24$ | $117 \pm 4.19$ |
| MFM $k_Q = 10^3$ | $2.67 \pm 2.16$ | $21.0 \pm 2.66$ | $53.6 \pm 1.33$ | 1.12e-1$\pm$.04 | $28.1 \pm 0.23$ | $143 \pm 14.5$ |
| DDS | $15.2 \pm 35.9$ | $18.0 \pm 36.9$ | $2400 \pm 8.65$ | 7.59e-2$\pm$.02 | $24.7 \pm 0.08$ | $3260 \pm 8.41$ |
| NF-MCMC | $548 \pm 325$ | $549 \pm 325$ | $2000 \pm 15.6$ | $11.8 \pm 7.55$ | $89.0 \pm 238$ | $215 \pm 46.4$ |
| FAB | $0.14 \pm 0.42$ | $1.78 \pm 0.42$ | $3880 \pm 7.19$ | 1.55e-1$\pm$.06 | $52.3 \pm 2.02$ | $1040 \pm 2.78$ |
| AT-SMC | $1.61 \pm 2.33$ | $18.4 \pm 2.35$ | $4.13 \pm 0.30$ | 1.39e-2$\pm$.01 | $25.0 \pm 0.12$ | $6.11 \pm 0.44$ |

Table 2: Diagnostics for the two real data examples. KSD U-stat and V-stat are the Kernel Stein Discrepancy U- and V-statistics between the target and samples generated from the learned flow. Results are averaged and empirical 95% confidence intervals over 10 independent runs.

## 6   Conclusion

**Summary**. In this paper, we introduced Markovian Flow Matching, a new approach to sampling from unnormalized probability distributions that augments MCMC with CNFs. Our method combines a local Markov kernel with a non-local, flow-informed Markov kernel, which is adaptively learned during sampling using FM. It also incorporates an adaptive tempering mechanism, which allows for the discovery of multiple target modes. Under mild assumptions, we established convergence of the flow network parameters output by our algorithm to a local optimum of the FM objective. We also benchmarked the performance of our algorithm on several examples, illustrating comparable performance to other state-of-the-art methods, often at a fraction of the computational cost.

**Limitations and Future Work**. We highlight three limitations of our work. First, our theoretical result established convergence of the flow network parameters obtained via MFM to a *local* minimum of the FM objective. Further work is required to understand how well these local minima generalize, in order to accurately quantify how accurately the corresponding CNF captures the target posterior. Second, we did not establish non-asymptotic convergence rates for our method. Finally, since it was not the main focus of this work, we did not explore in great detail other choices of architecture for the flow network. We expect that, for certain targets, this could have a significant impact on the performance of MFM. Indeed, a promising avenue for further research lies in developing tailored CNFs designed for particular posterior distributions. This approach would go beyond the current practice of including the gradient of the log-posterior and instead exploit unique characteristics intrinsic to each model when constructing the flow.

## Acknowledgments and Disclosure of Funding

LS and CN were supported by the Engineering and Physical Sciences Research Council (EPSRC), grant number EP/V022636/1. CN acknowledges further support from the EPSRC, grant numbers EP/S00159X/1 and EP/Y028783/1.

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

# A    Flow-Informed Markov Chain Monte Carlo Methods

---

**Algorithm 2** Flow-informed Random Walk Metropolis Hastings

---

1: **Input:** initial $x$, target $\pi$, vector field $v_t^\theta$, flow parameters $\theta$
2: **Output:** $x'$
3: $\sigma_{opt} \leftarrow 2.38/\sqrt{d}$
4: $\phi_1(x) = x_{t=1} \leftarrow x$
5: $\begin{bmatrix} x_0 \\ \Delta \log p(x_0) \end{bmatrix} \leftarrow \begin{bmatrix} x \\ 0 \end{bmatrix} + \int_1^0 \begin{bmatrix} v_t^\theta(\phi_t(x)) \\ -\nabla \cdot v_t^\theta(\phi_t(x)) \end{bmatrix} \mathrm{d}t$
6: $y_0 \sim \mathcal{N}(\cdot | x_0, \sigma_{opt}^2)$
7: $\begin{bmatrix} y_1 \\ \Delta \log p(y_1) \end{bmatrix} \leftarrow \begin{bmatrix} y_0 \\ 0 \end{bmatrix} + \int_0^1 \begin{bmatrix} v_t^\theta(\phi_t(y)) \\ -\nabla \cdot v_t^\theta(\phi_t(y)) \end{bmatrix} \mathrm{d}t$
8: $\alpha \leftarrow \min \left\{ 1, \frac{\pi(y_1) \exp(-\Delta \log p(y_1))}{\pi(x_1) \exp(\Delta \log p(x_0))} \right\}$
9: With probability $\alpha$ make $x' \leftarrow y_1$ else $x' \leftarrow x$

---

---

**Algorithm 3** Flow-informed Independent Metropolis Hastings

---

1: **Input:** initial $x$, target density $\pi$, vector field $v_t^\theta$, reference density $p_0$, flow parameters $\theta$.
2: **Output:** $x'$
3: $\phi_1(u) = u_{t=1} \leftarrow x$
4: $\begin{bmatrix} u_0 \\ \Delta \log p(u_0) \end{bmatrix} \leftarrow \begin{bmatrix} u_1 \\ 0 \end{bmatrix} + \int_1^0 \begin{bmatrix} v_t^\theta(\phi_t(u)) \\ -\nabla \cdot v_t^\theta(\phi_t(u)) \end{bmatrix} dt$
5: $\phi_0(x) = x_{t=0} \sim p_0$
6: $\begin{bmatrix} x_1 \\ \log p(x_1) \end{bmatrix} \leftarrow \begin{bmatrix} x_0 \\ \log p_0(x_0) \end{bmatrix} + \int_0^1 \begin{bmatrix} v_t^\theta(\phi_t(x)) \\ -\nabla \cdot v_t^\theta(\phi_t(x)) \end{bmatrix} dt$
7: $\alpha \leftarrow \min \left\{ 1, \frac{\pi(x_1) p_0(u_0) \exp(-\Delta \log p(u_0))}{\exp(\log p(x_1)) \pi(u_1)} \right\}$
8: With probability $\alpha$ make $x' \leftarrow x_1$ else $x' \leftarrow x$

---

---

**Algorithm 4** Flow-informed Conditional Importance Sampling

---

1: **Input:** initial $x$, target density $\pi$, vector field $v_t^\theta$, reference density $q_0$, flow parameters $\theta$, number of importance samples $K$.
2: **Output:** $x'$
3: $\phi_1(u) = u_{t=1} \leftarrow x$
4: $\begin{bmatrix} u_0 \\ \Delta \log p(u_0) \end{bmatrix} \leftarrow \begin{bmatrix} u_1 \\ 0 \end{bmatrix} + \int_1^0 \begin{bmatrix} v_t^\theta(\phi_t(u)) \\ -\nabla \cdot v_t^\theta(\phi_t(u)) \end{bmatrix} dt$
5: $w_0 \leftarrow \frac{\pi(u_1)}{p_0(u_0) \exp(-\Delta \log p(u_0))}$
6: $x^{(0)} \leftarrow x$
7: **for** $k = 1 : K$ **do**
8:      $\phi_0(x) = x_{t=0} \sim q_0$
9:      $\begin{bmatrix} x_1 \\ \log p(x_1) \end{bmatrix} \leftarrow \begin{bmatrix} x_0 \\ \log p_0(x_0) \end{bmatrix} + \int_0^1 \begin{bmatrix} v_t^\theta(\phi_t(x)) \\ -\nabla \cdot v_t^\theta(\phi_t(x)) \end{bmatrix} dt$
10:      $w_k \leftarrow \frac{\pi(x_1)}{\exp(\log p(x_1))}$
11:      $x^{(k)} \leftarrow x_1$
12: **end for**
13: Choose $k'$ with probability $P(k' = k) \propto w_k$, then make $x' \leftarrow x^{(k')}$

---

# B  Proof of Proposition 3.1

Our proof follows closely the proof of [54, Proposition 1]. Let $\theta^*$ be a minimizer of the CFM objective in (4), which we recall is given by

$$\mathcal{J}(\theta;\pi) = \mathbb{E}_{x_1 \sim \pi}\mathbb{E}_{t \sim \mathcal{U}(0,1)}\mathbb{E}_{x \sim p_t(\cdot|x_1)}\left[||v_t^\theta(x) - v_t(x|x_1)||^2\right] := \mathbb{E}_{x_1 \sim \pi}\left[j(\theta,x_1)\right] \quad (20)$$

where we have defined $j(\theta,x_1) = \mathbb{E}_{t \sim \mathcal{U}(0,1)}\mathbb{E}_{x \sim p_t(\cdot|x_1)}\left[||v_t^\theta(x) - v_t(x|x_1)||^2\right]$. Now, consider the ordinary differential equation (ODE) given by

$$\frac{\mathrm{d}}{\mathrm{d}t}\theta(t) = \nabla\mathcal{J}_\theta(\theta(t);\pi), \quad \theta(0) = \theta_0, \quad t \geq 0. \quad (21)$$

We say that $\hat{\theta}$ is stability point of this ODE if, given the initial condition $\theta(0) = \hat{\theta}$, the ODE admits the unique solution $\theta(t) = \hat{\theta}$ for all $t \geq 0$. Naturally, the minimizer $\theta^*$ is a stability point of this ODE, since $\nabla_\theta\mathcal{J}(\theta;\pi)|_{\theta=\theta^*} = 0$. Meanwhile, we call $\Theta$ the domain of attraction of $\theta^*$ if, given the initial condition $\theta(0) \in \Theta$, the solution $\theta(t) \in \Theta$ for all $t \geq 0$, and $\theta(t)$ converges to $\theta^*$ as $t \to \infty$.

Let $x_k \in \mathbb{R}^{d_x}$, and $\mathcal{X} \subseteq \mathbb{R}^{d_x}$ be an open subset of $\mathbb{R}^{d_x}$. Let $\Theta \subseteq \mathbb{R}^{d_\theta}$ be an open set in $\mathbb{R}^{d_\theta}$, and $\Theta_c \subseteq \Theta$ be a compact subset of $\Theta$. Consider the Markov transition kernel $M := P \circ Q^k$ given by a cycle of $k_Q$ repeated transitions of a MALA transition kernel and a flow-informed RWMH transition kernel, viz

$$M_{\pi,\theta}(x,\mathrm{d}y) = \int \cdots \int Q(x,\mathrm{d}x_1;\theta)Q(x_1,\mathrm{d}x_2)\ldots Q(x_{k_Q-1},\mathrm{d}x_{k_Q};\theta)P(x_{k_Q},\mathrm{d}y;\pi,\theta). \quad (22)$$

This transition kernel is $\pi$-invariant since both $P$ and $Q$ are $\pi$-invariant. In addition, let $M_{\pi,\theta}^k(x,\mathrm{d}y)$ be the repeated application of this Markov transition kernel, namely,

$$M_{\pi,\theta}^k(x,\mathrm{d}y) = \int \cdots \int M_{\pi,\theta}(x,\mathrm{d}x_1)M_{\pi,\theta}(x_1,\mathrm{d}x_2)\cdots M_{\pi,\theta}(x_{k-2},\mathrm{d}x_{k-1})M_{\pi,\theta}(x_{k-1},\mathrm{d}y). \quad (23)$$

Following [30, 54], we impose the following assumptions, for some sufficiently large positive real number $q > 1$.

**Assumption B.1** (Robbins-Monro Condition). The step size sequence $(\varepsilon_k)_{k=1}^\infty$ satisfies the following requirements:

$$\sum_{k=1}^\infty \varepsilon_k = \infty, \quad \sum_{k=1}^\infty \varepsilon_k^2 < \infty. \quad (24)$$

**Assumption B.2** (Integrability). There exists a constant $C_1 > 0$ such that for, any $\theta \in \Theta$, $x \in \mathcal{X}$ and $k \geq 1$,

$$\int (1 + |y|^q)M_{\pi,\theta}^k(x,\mathrm{d}y) \leq C_1(1 + |x|^q). \quad (25)$$

**Assumption B.3** (Convergence of the Markov Chain). For each $\theta \in \Theta$, it holds that

$$\lim_{k \to \infty} \sup_{x \in \mathcal{X}} \frac{1}{1 + |x|^q} \int (1 + |y|^q)|M_{\pi,\theta}^k(x,\mathrm{d}y) - \pi(\mathrm{d}y)| = 0. \quad (26)$$

**Assumption B.4** (Continuity in $\theta$). There exists a constant $C_2$ such that for all $\theta,\theta' \in \Theta_c$,

$$\left|\int (1 + |y|^q)(M_{\pi,\theta}^k(x,\mathrm{d}y) - M_{\pi,\theta'}^k(x,\mathrm{d}y))\right| \leq C_2|\theta - \theta'|(1 + |x|^q). \quad (27)$$

**Assumption B.5** (Continuity in $x$). There exists a constant $C_3$ such that for all $x_1, x_2 \in \mathcal{X}$,

$$\sup_{\theta \in \Theta}\left|\int (1 + |y|^{q+1})(M_{\pi,\theta}^k(x_1,\mathrm{d}y) - M_{\pi,\theta}^k(x_2,\mathrm{d}y))\right| \leq C_3|x_1 - x_2|(1 + |x_1|^q + |x_2|^q). \quad (28)$$

**Assumption B.6** (Conditions on the Objective Function). For any compact subset $\Theta_c \subset \Theta$, there exist positive constants $p, K_1, K_2, K_3$ and $v > 1/2$ such that for all $\theta, \theta' \in \Theta_c$ and $x, x_1, x_2 \in \mathcal{X}$,

$$|\nabla_\theta j(\theta,x_1)| \leq K_1(1 + |x_1|^{p+1}), \quad (29)$$

$$|\nabla_\theta j(\theta,x_1) - \nabla_\theta j(\theta,x_1')| \leq K_2|x_1 - x_1'|(1 + |x_1|^p + |x_2|^p), \quad (30)$$

$$|\nabla_\theta j(\theta,x_1) - \nabla_\theta j(\theta',x_1)| \leq K_3|\theta - \theta'|^v(1 + |x_1|^{p+1}). \quad (31)$$

With the above assumptions, the result follows from Theorem 1 of [30] by setting $x \to x_1$, $\Pi_\theta \to M_{\pi,\theta}$, $H(\theta,x) \to \nabla_\theta j(\theta,x_1)$.

## C Additional Experimental Details

Code for the numerical experiments is written in Python with array computations handled by JAX [11]. The implementation of relevant methods for comparison is sourced from open source repositories: DDS using franciscovargas/denoising_diffusion_samplers, NF-MCMC using kazewong/flowMC [80], and FAB using lollcat/fab-jax [52]. All experiments are run on an NVIDIA V100 GPU with 32GB of memory. In the following subsections, we will give more details on the modelling and hyperparameter choices for each experiment, along with additional results.

### C.1 Diagnostics

Let $\pi$ and $\nu$ be two probability measures. Let $\mathcal{F}$ denote the unit ball in a reproducing kernel Hilbert space (RKHS) $\mathcal{H}$, associated with the positive definite kernel $k : \mathbb{R}^d \times \mathbb{R}^d \to \mathbb{R}$. Then the maximum mean discrepancy (MMD) between $\pi$ and $\nu$ is defined as [29, Section 2.2]

$$\mathrm{MMD}_k^2(\pi, \nu) = \|m_\pi - m_\nu\|_\mathcal{F}^2, \tag{32}$$

where $m_\pi$ is the mean embedding of $\pi$, defined via $\mathbb{E}_\pi[f] = \langle f, m_\pi \rangle_\mathcal{H}$ for all $f \in \mathcal{H}$. Using standard properties of the RKHS, the squared MMD can be written as [29, Lemma 6]

$$\mathrm{MMD}_k^2(\pi, \nu) = \mathbb{E}_{x,x'\sim\pi}[k(x,x')] - 2\mathbb{E}_{x\sim\pi,y\sim\nu}[k(x,y)] + \mathbb{E}_{y\sim\nu,y'\sim\nu}[k(y,y')]. \tag{33}$$

Thus, given samples $(x_i)_{i=1}^m \sim \pi$ and $(y_i)_{i=1}^m \sim \nu$, an unbiased estimate of the squared MMD can be computed as

$$\widehat{\mathrm{MMD}}_k^2(\pi, \nu) = \frac{1}{m(m-1)} \sum_{i=1}^m \sum_{i\neq j}^m k(x_i, x_j) - \frac{2}{m^2} \sum_{i=1}^m \sum_{j=1}^m k(x_i, y_j)$$

$$+ \frac{1}{m(m-1)} \sum_{i=1}^m \sum_{j\neq i}^m k(y_i, y_j). \tag{34}$$

For a kernel $k$, the kernel Stein discrepancy (KSD) between $\pi$ and $\nu$ is defined as the MMD between $\pi$ and $\nu$, using the Stein kernel $k_\pi$ associated with $k$, which is defined as

$$k_\pi(x, x') = \nabla_x \cdot \nabla_{x'} k(x, x') + \nabla_x k(x, x') \cdot \nabla_{x'} \log \pi(x')$$
$$+ \nabla_{x'} k(x, x') \cdot \nabla_x \log \pi(x) + k(x, x')\nabla_x \log \pi(x) \cdot \nabla_{x'} \log \pi(x), \tag{35}$$

and satisfies the Stein identity $\mathbb{E}_\pi[k_\pi(x, \cdot)] = 0$. We thus have that

$$\mathrm{KSD}_k^2(\pi, \nu) = \mathrm{MMD}_{k_\pi}^2(\pi, \nu) \tag{36}$$

$$= \mathbb{E}_{x,x'\sim\pi}[k_\pi(x,x')] - 2\mathbb{E}_{x\sim\pi,y\sim\nu}[k_\pi(x,y)] + \mathbb{E}_{y\sim\nu,y'\sim\nu}[k_\pi(y,y')] \tag{37}$$

$$= \mathbb{E}_{y\sim\nu,y'\sim\nu}[k_\pi(y,y')]. \tag{38}$$

We can obtain estimates of the KSD by using U-statistics or V-statistics. In particular, an unbiased estimate of $\mathrm{KSD}_k^2(\pi, \nu)$ is given by the U-statistic [42]

$$\widehat{\mathrm{KSD}}_{k,U}^2(\pi, \nu) = \frac{1}{n(n-1)} \sum_{i=1}^n \sum_{i\neq j}^n k_\pi(y_i, y_i'). \tag{39}$$

Alternatively, we can estimate $\mathrm{KSD}_k^2(\pi, \nu)$ using a biased (but non-negative) V-statistic of the form [46, Section 4]

$$\widehat{\mathrm{KSD}}_{k,V}^2(\pi, \nu) = \frac{1}{n^2} \sum_{i=1}^n \sum_{j=1}^n k_\pi(y_i, y_i'). \tag{40}$$

In all of our numerical experiments, we calculate the U- and V- statistics using the inverse multi-quadratic kernel $k(x, x') = (1 + (x - x')^T(x - x'))^\beta$ due to its favourable convergence properties [26, Theorem 8], setting $\beta = -\frac{1}{2}$.

## C.2 4-mode Gaussian mixture

For this experiment, all methods use $N = 128$ parallel chains for training and 128 hidden dimensions for all neural networks. Methods with a MALA kernel use a step size of 0.2, and methods with splines use 4 coupling layers with 8 bins and range limited to $[-16, 16]$.

In Table 3, we present results for $K = 10^3$ iterations. Since MFM is much more efficient than other methods, we also report results for a great number of total iterations. Table 4 contains results for $K = 5 \cdot 10^3$ iterations for MFM and AT-SMC. In the main text, we present results for $K = 5 \cdot 10^3$ learning iterations for MFM and AT-SMC, and $K = 10^3$ iterations for the other algorithms, since this renders the total computational cost of all algorithms somewhat comparable.

For both choices of $K$, we also present results using Hutchinson's trace estimator (HTE) [27, 36] to calculate the MH acceptance probability in the flow-informed Markov transition kernel. As expected, its effect on sample quality becomes more apparent as $k_Q$ increases. However, its effect on computation time is less significant than in larger dimensional examples.

| $K = 10^3$ | $\mathbb{E}_{[\phi_1]_{\#}p_0} \log \pi$ | KSD U-stat. | KSD V-stat. | MMD | seconds |
|---|---|---|---|---|---|
| FM w/ $\pi$ samples | $-4.20 \pm 0.08$ | 6.85e-3$\pm$3.75e-3 | 7.16e-3$\pm$3.75e-3 | 1.90e-3$\pm$1.16e-3 | $6.46 \pm 0.32$ |
| MFM $k_Q = K$ | $-4.55 \pm 0.16$ | 7.62e-3$\pm$7.46e-3 | 7.98e-3$\pm$7.45e-3 | 3.00e-3$\pm$2.20e-3 | $10.4 \pm 0.66$ |
| MFM $k_Q = 10^2$ | $-4.50 \pm 0.14$ | 5.36e-3$\pm$3.18e-3 | 5.71e-3$\pm$3.19e-3 | 2.03e-3$\pm$2.12e-3 | $12.1 \pm 0.65$ |
| – w/ HTE | $-4.52 \pm 0.15$ | 4.77e-3$\pm$1.93e-3 | 5.12e-3$\pm$1.93e-3 | 2.27e-3$\pm$1.30e-3 | $13.9 \pm 0.33$ |
| MFM $k_Q = 10$ | $-4.49 \pm 0.08$ | 7.01e-3$\pm$3.49e-3 | 7.36e-3$\pm$3.49e-3 | 1.62e-3$\pm$7.61e-4 | $24.8 \pm 1.38$ |
| – w/ HTE | $-4.53 \pm 0.12$ | 1.10e-2$\pm$6.80e-3 | 1.14e-2$\pm$6.81e-3 | 2.64e-3$\pm$1.03e-3 | $30.0 \pm 1.70$ |
| DDS | $-4.22 \pm 0.03$ | 9.89e-4$\pm$1.05e-3 | 1.30e-3$\pm$1.05e-3 | 1.76e-4$\pm$2.32e-4 | $114. \pm 0.68$ |
| NF-MCMC | $-4.37 \pm 0.21$ | 1.80e-2$\pm$1.44e-2 | 1.83e-2$\pm$1.44e-2 | 5.85e-3$\pm$3.91e-3 | $72.0 \pm 11.7$ |
| FAB | $-4.67 \pm 0.16$ | 2.31e-3$\pm$1.19e-3 | 2.69e-3$\pm$1.21e-3 | 2.69e-4$\pm$2.06e-4 | $101. \pm 3.24$ |
| AT-SMC | $-4.47 \pm 0.04$ | 3.95e-3$\pm$2.06e-3 | 4.30e-3$\pm$2.06e-3 | 2.98e-2$\pm$4.08e-2 | $1.38 \pm 0.08$ |

Table 3: Diagnostics for the 4-mode Gaussian mixture with $K = 10^3$. $\mathbb{E}_{[\phi_1]_{\#}p_0} \log \pi$ is the Monte Carlo approximation of the log-target density using the learned flow to generate samples; KSD U-stat and V-stat are the Kernel Stein Discrepancy U- and V-statistics between the target and samples generated from the learned flow; MMD is the Maximum Mean Discrepancy between real samples from the target and samples generated from the learned flow. Results are averaged and empirical 95% confidence intervals over 10 independent runs.

| $K = 5 \cdot 10^3$ | $\mathbb{E}_{[\phi_1]_{\#}p_0} \log \pi$ | KSD U-stat. | KSD V-stat. | MMD | seconds |
|---|---|---|---|---|---|
| FM w/ $\pi$ samples | $-4.22 \pm 0.04$ | 1.50e-3$\pm$6.38e-4 | 1.81e-3$\pm$6.33e-4 | 3.69e-4$\pm$1.84e-4 | $22.3 \pm 0.64$ |
| MFM $k_Q = K$ | $-4.47 \pm 0.05$ | 3.15e-3$\pm$2.10e-3 | 3.50e-3$\pm$2.10e-3 | 2.37e-3$\pm$2.29e-3 | $27.9 \pm 1.27$ |
| MFM $k_Q = 10^2$ | $-4.45 \pm 0.04$ | 3.61e-3$\pm$2.07e-3 | 3.96e-3$\pm$2.07e-3 | 1.05e-3$\pm$8.90e-4 | $39.2 \pm 1.74$ |
| – w/ HTE | $-4.48 \pm 0.09$ | 3.50e-3$\pm$2.22e-3 | 3.86e-3$\pm$2.22e-3 | 1.88e-3$\pm$1.96e-3 | $41.2 \pm 1.65$ |
| MFM $k_Q = 10$ | $-4.44 \pm 0.07$ | 3.15e-3$\pm$2.28e-3 | 3.49e-3$\pm$2.28e-3 | 8.13e-4$\pm$4.41e-4 | $117. \pm 5.65$ |
| – w/ HTE | $-4.46 \pm 0.06$ | 4.80e-3$\pm$3.17e-3 | 5.15e-3$\pm$3.17e-3 | 1.37e-3$\pm$9.65e-4 | $147. \pm 9.44$ |
| AT-SMC | $-4.48 \pm 0.04$ | 4.07e-3$\pm$1.24e-3 | 4.42e-3$\pm$1.24e-3 | 3.95e-2$\pm$2.90e-2 | $2.18 \pm 0.26$ |

Table 4: Diagnostics for the 4-mode Gaussian mixture with $K = 5 \cdot 10^3$. $\mathbb{E}_{[\phi_1]_{\#}p_0} \log \pi$ is the Monte Carlo approximation of the log-target density using the learned flow to generate samples; KSD U-stat and V-stat are the Kernel Stein Discrepancy U- and V-statistics between the target and samples generated from the learned flow; MMD is the Maximum Mean Discrepancy between real samples from the target and samples generated from the learned flow. Results are averaged and empirical 95% confidence intervals over 10 independent runs.

## C.3 16-mode Gaussian Mixture

Like the 4-mode example, all methods use $N = 128$ parallel chains for training and 128 hidden dimensions for all neural networks. Methods with a MALA kernel use a step size of 0.2, and methods with splines use 4 coupling layers with 8 bins and range limited to $[-16, 16]$. In Table 5 we present results for $K = 10^3$ iterations. In Table 6, we provide results for MFM and AT-SMC for $K = 5 \times 10^3$ learning iterations. In the main text, we present results for $K = 5 \cdot 10^3$ learning iterations for MFM and AT-SMC and $K = 10^3$ iterations for all other algorithms, which yields a more comparable total computation time.

| $K = 10^3$ | $\mathbb{E}_{[\phi_1]_{\#}p_0} \log \pi$ | KSD U-stat. | KSD V-stat. | MMD | seconds |
|---|---|---|---|---|---|
| FM w/ $\pi$ samples | $-7.09 \pm 0.31$ | 2.94e-2±1.32e-2 | 2.99e-2±1.32e-2 | 8.89e-3±1.79e-3 | $6.67 \pm 0.19$ |
| MFM $k_Q = K$ | $-6.95 \pm 0.47$ | 1.67e-2±1.14e-2 | 1.71e-2±1.15e-2 | 3.71e-2±4.81e-3 | $10.4 \pm 0.64$ |
| MFM $k_Q = 10^2$ | $-7.21 \pm 0.73$ | 1.80e-2±9.31e-3 | 1.85e-2±9.43e-3 | 1.81e-2±8.97e-3 | $11.4 \pm 0.74$ |
| – w/ HTE | $-7.34 \pm 0.81$ | 2.10e-2±8.65e-3 | 2.15e-2±8.75e-3 | 1.74e-2±8.12e-3 | $13.0 \pm 0.83$ |
| MFM $k_Q = 10$ | $-7.21 \pm 0.58$ | 2.95e-2±1.03e-2 | 3.01e-2±1.04e-2 | 1.06e-2±2.76e-3 | $20.3 \pm 1.17$ |
| – w/ HTE | $-7.18 \pm 0.85$ | 3.17e-2±1.97e-2 | 3.23e-2±1.99e-2 | 1.30e-2±3.33e-3 | $22.5 \pm 1.81$ |
| DDS | $-5.86 \pm 0.20$ | 6.65e-3±6.69e-3 | 6.94e-3±6.69e-3 | 1.02e-1±4.10e-2 | $115. \pm 0.64$ |
| NF-MCMC | $-5.74 \pm 0.35$ | 1.23e-2±1.71e-2 | 1.26e-2±1.72e-2 | 8.05e-3±1.42e-2 | $67.0 \pm 12.3$ |
| FAB | $-5.89 \pm 0.28$ | 4.22e-3±3.31e-3 | 4.58e-3±3.34e-3 | 1.51e-3±1.06e-3 | $102. \pm 4.32$ |
| AT-SMC | $-5.91 \pm 0.07$ | 2.07e-3±1.03e-3 | 2.38e-3±1.04e-3 | 3.72e-2±4.45e-3 | $1.36 \pm 0.20$ |

Table 5: Diagnostics for the 16-mode Gaussian mixture with $K = 10^3$. $\mathbb{E}_{[\phi_1]_{\#}p_0} \log \pi$ is the Monte Carlo approximation of the log-target density using the learned flow to generate samples; KSD U-stat and V-stat are the Kernel Stein Discrepancy U- and V-statistics between the target and samples generated from the learned flow; MMD is the Maximum Mean Discrepancy between real samples from the target and samples generated from the learned flow. Results are averaged and empirical 95% confidence intervals over 10 independent runs.

| $K = 5 \cdot 10^3$ | $\mathbb{E}_{[\phi_1]_{\#}p_0} \log \pi$ | KSD U-stat. | KSD V-stat. | MMD | seconds |
|---|---|---|---|---|---|
| FM w/ $\pi$ samples | $-5.74 \pm 0.11$ | 2.91e-3±1.23e-3 | 3.26e-3±1.24e-3 | 1.35e-3±6.66e-4 | $22.4 \pm 1.01$ |
| MFM $k_Q = K$ | $-6.09 \pm 0.10$ | 3.00e-3±7.87e-4 | 3.34e-3±7.95e-4 | 1.88e-2±3.67e-3 | $28.2 \pm 2.84$ |
| MFM $k_Q = 10^2$ | $-5.90 \pm 0.08$ | 5.37e-3±2.00e-3 | 5.74e-3±2.01e-3 | 2.98e-3±1.37e-3 | $34.8 \pm 1.95$ |
| – w/ HTE | $-5.88 \pm 0.12$ | 5.43e-3±2.32e-3 | 5.80e-3±2.33e-3 | 3.86e-3±1.23e-3 | $38.7 \pm 2.53$ |
| MFM $k_Q = 10$ | $-5.98 \pm 0.13$ | 5.48e-3±2.07e-3 | 5.86e-3±2.09e-3 | 2.87e-3±9.67e-4 | $89.6 \pm 5.19$ |
| – w/ HTE | $-5.92 \pm 0.09$ | 9.18e-3±4.48e-3 | 9.57e-3±4.50e-3 | 8.58e-3±1.00e-3 | $110. \pm 6.77$ |
| AT-SMC | $-5.84 \pm 0.05$ | 2.09e-3±8.53e-4 | 2.40e-3±8.56e-4 | 1.73e-2±5.30e-3 | $2.19 \pm 0.21$ |

Table 6: Diagnostics for the 16-mode Gaussian mixture with $K = 5 \cdot 10^3$. $\mathbb{E}_{[\phi_1]_{\#}p_0} \log \pi$ is the Monte Carlo approximation of the log-target density using the learned flow to generate samples; KSD U-stat and V-stat are the Kernel Stein Discrepancy U- and V-statistics between the target and samples generated from the learned flow; MMD is the Maximum Mean Discrepancy between real samples from the target and samples generated from the learned flow. Results are averaged and empirical 95% confidence intervals over 10 independent runs.

## C.4 Many Well

We also present a synthetic problem approximating the 32-dimensional Many Well distribution given by the product of 16 copies of the 2-dimensional Double Well distribution [e.g., 52, 58, 81],

$$\log p(x_1, x_2) = -x_1^4 + 6x_1^2 + \frac{1}{2}x_1 - \frac{1}{2}x_2^2 + \text{constant}, \tag{41}$$

where each copy of the Double Well is evaluated on a different pair of the 32 inputs. The 32-dimensional Many Well has $2^{16} = 65536$ modes, one for each possible choice of mode in each of the 16 copies of the double well. We can obtain exact samples from the Many Well by sampling each independent copy of the Double Well.

Like previous synthetic examples, all methods use $N = 128$ parallel chains for training and 128 hidden dimensions for all neural networks. Methods with a MALA kernel use a step size of 0.1, and methods with splines use 4 coupling layers with 8 bins and a range limited to $[-16, 16]$. Table 7 presents results for $K = 10^3$ iterations. In Table 8, we provide results for MFM and AT-SMC for $K = 5 \times 10^3$ learning iterations. In the main text, we present results for $K = 5 \cdot 10^3$ learning iterations for MFM and AT-SMC and $K = 10^3$ iterations for all other algorithms, which yields a more comparable total computation time.

## C.5 Field system

The stochastic Allen–Cahn equation is defined in terms of a random field $\phi : [0, 1] \to \mathbb{R}$ satisfying the following stochastic partial differential equation [e.g., 24, Section V]:

$$\frac{\partial \phi}{\partial t} = a \frac{\partial^2 \phi}{\partial s^2} + a^{-1}(\phi - \phi^3) + \sqrt{2\beta^{-1}}\eta(t, s), \tag{42}$$

| $K = 10^3$ | $\mathbb{E}_{[\phi_1]_{\#}p_0} \log \pi$ | KSD U-stat. | KSD V-stat. | MMD | seconds |
|---|---|---|---|---|---|
| FM w/ $\pi$ samples | $86.6 \pm 1.88$ | $17.4 \pm 6.24$ | $19.3 \pm 6.25$ | 1.24e-8±1.81e-8 | $6.95 \pm 0.31$ |
| MFM $k_Q = K$ | $101. \pm 0.70$ | $0.12 \pm 0.12$ | $0.77 \pm 0.13$ | 2.28e-8±1.57e-8 | $11.1 \pm 0.67$ |
| MFM $k_Q = 10^2$ | $101. \pm 0.70$ | $0.12 \pm 0.11$ | $0.77 \pm 0.13$ | 2.28e-8±1.57e-8 | $120. \pm 17.6$ |
| – w/ HTE | $101. \pm 0.70$ | $0.11 \pm 0.12$ | $0.77 \pm 0.13$ | 2.28e-8±1.57e-8 | $35.6 \pm 3.95$ |
| MFM $k_Q = 10$ | $101. \pm 0.77$ | $0.11 \pm 0.11$ | $0.76 \pm 0.12$ | 2.28e-8±1.57e-8 | $1100. \pm 90.2$ |
| – w/ HTE | $101. \pm 0.69$ | $0.11 \pm 0.09$ | $0.76 \pm 0.10$ | 2.27e-8±1.56e-8 | $259. \pm 17.6$ |
| DDS | $133. \pm 3.92$ | $0.13 \pm 0.14$ | $0.61 \pm 0.13$ | 1.49e-5±2.03e-5 | $227. \pm 0.91$ |
| NF-MCMC | $39.2 \pm 1.61$ | $1.17 \pm 0.65$ | $2.29 \pm 0.66$ | 4.79e-7±2.41e-7 | $184. \pm 44.1$ |
| FAB | $137. \pm 0.33$ | 4.79e-3±2.55e-2 | 4.32e-1±4.14e-2 | 3.87e-7±2.90e-7 | $304. \pm 3.64$ |
| AT-SMC | $130. \pm 2.93$ | $0.02 \pm 0.03$ | $0.57 \pm 0.03$ | 1.75e-5±1.19e-5 | $1.35 \pm 0.36$ |

Table 7: Diagnostics for the many well with $K = 10^3$. $\mathbb{E}_{[\phi_1]_{\#}p_0} \log \pi$ is the Monte Carlo approximation of the log-target density using the learned flow to generate samples; KSD U-stat and V-stat are the Kernel Stein Discrepancy U- and V-statistics between the target and samples generated from the learned flow; MMD is the Maximum Mean Discrepancy between real samples from the target and samples generated from the learned flow. Results are averaged and empirical 95% confidence intervals over 10 independent runs.

| $K = 5 \cdot 10^3$ | $\mathbb{E}_{[\phi_1]_{\#}p_0} \log \pi$ | KSD U-stat. | KSD V-stat. | MMD | seconds |
|---|---|---|---|---|---|
| FM w/ $\pi$ samples | $98.6 \pm 5.32$ | $2.57 \pm 4.07$ | $4.56 \pm 4.07$ | −1.13e-9±2.81e-8 | $25.6 \pm 0.48$ |
| MFM $k_Q = K$ | $101. \pm 0.64$ | 1.02e-1±9.03e-2 | 7.64e-1±9.78e-2 | 2.28e-8±1.57e-8 | $29.7 \pm 1.03$ |
| MFM $k_Q = 10^2$ | $101. \pm 0.63$ | 1.02e-1±9.05e-2 | 7.64e-1±9.73e-2 | 2.28e-8±1.57e-8 | $587. \pm 27.0$ |
| – w/ HTE | $102. \pm 0.78$ | 1.02e-1±8.32e-2 | 7.62e-1±8.91e-2 | 2.27e-8±1.58e-8 | $154. \pm 7.51$ |
| MFM $k_Q = 10$ | $102. \pm 0.81$ | 9.93e-2±7.24e-2 | 7.63e-1±8.15e-2 | 2.27e-8±1.57e-8 | $5130. \pm 104.$ |
| – w/ HTE | $103. \pm 2.20$ | 3.85e-1±2.69e-1 | $1.05 \pm 0.27$ | 2.21e-8±1.57e-8 | $1190. \pm 27.6$ |
| AT-SMC | $130. \pm 3.17$ | 2.44e-2±5.99e-2 | 5.79e-1±6.76e-2 | 1.52e-5±1.00e-5 | $2.59 \pm 0.27$ |

Table 8: Diagnostics for the many well with $K = 5 \cdot 10^3$. $\mathbb{E}_{[\phi_1]_{\#}p_0} \log \pi$ is the Monte Carlo approximation of the log-target density using the learned flow to generate samples; KSD U-stat and V-stat are the Kernel Stein Discrepancy U- and V-statistics between the target and samples generated from the learned flow; MMD is the Maximum Mean Discrepancy between real samples from the target and samples generated from the learned flow. Results are averaged and empirical 95% confidence intervals over 10 independent runs.

where $a > 0$ is a parameter, $\beta$ is the inverse temperature, $s \in [0, 1]$ denotes the spatial variable, and $\eta$ is spatiotemporal white noise. We impose Dirichlet boundary conditions throughout, so that $\phi(s = 0) = \phi(s = 1) = 0$.

The associated Hamiltonian, reflecting a spatial coupling term penalizing changes in $\phi$, takes the form:

$$U_*[\phi] = \beta \int_0^1 \left[ \frac{a}{2} \left( \frac{\partial \phi}{\partial s} \right)^2 + \frac{1}{4a} \left( 1 - \phi^2(s) \right)^2 \right] ds. \tag{43}$$

At low temperatures, this coupling induces alignment of the field in either the positive or negative direction, leading to two global minima, $\phi_+$ and $\phi_-$, with typical values of $\pm 1$.

For this example, all methods use $N = 1024$ parallel chains for training and 256 hidden dimensions for all neural networks. Methods with a MALA kernel use a step size of $0.0001$, and methods with splines use 8 coupling layers with 8 bins and range limited to $[-5, 5]$. Results for $K = 10^4$ learning iterations are presented in Table 9. In the main text, we present results for $k_Q = 10^2$ for MFM.

Interestingly, in high-dimensional problems such as this and the following, the version of the algorithm using Hutchinson's trace estimator [27, 36] to calculate the MH acceptance probability has little apparent effect on the approximation quality. It does, however, have a significant impact on the computation time.

## C.6 Log-Gaussian Cox process

The original $10 \times 10$ square meter plot is standardized to the unit square. We discretize the unit square $[0, 1]^2$ into a $M = 40 \times 40$ regular grid. The latent intensity process $\Lambda = \{\Lambda_m\}_{m \in M}$ is specified as $\Lambda_m = \exp(X_m)$, where $X = \{X_m\}_{m \in M}$ is a Gaussian process with a constant mean $\mu_0 \in \mathbb{R}$ and exponential covariance function $\Sigma_0(m, n) = \sigma^2 \exp\left(-|m - n|/(40\beta)\right)$ for $m, n \in M$,

| $K = 10^4$ | $\mathbb{E}_{[\phi_1]_{\#}p_0} \log \pi$ | KSD U-stat. | KSD V-stat. | seconds |
|---|---|---|---|---|
| MFM $k_Q = K$ | $-74.7 \pm 5.67$ | $2.61 \pm 2.00$ | $20.9 \pm 2.49$ | $52.3 \pm 1.23$ |
| MFM $k_Q = K/10$ | $-69.6 \pm 6.07$ | $2.90 \pm 2.50$ | $21.2 \pm 3.16$ | $61.7 \pm 4.37$ |
| – w/ HTE | $-74.2 \pm 6.51$ | $2.67 \pm 2.16$ | $21.0 \pm 2.66$ | $53.6 \pm 1.33$ |
| MFM $k_Q = K/100$ | $-55.4 \pm 4.19$ | $2.84 \pm 3.31$ | $20.9 \pm 3.33$ | $180 \pm 42.2$ |
| – w/ HTE | $-72.1 \pm 11.6$ | $3.84 \pm 3.04$ | $22.6 \pm 4.26$ | $71.5 \pm 6.93$ |
| DDS | $-76.3 \pm 17.4$ | $15.2 \pm 35.9$ | $18.0 \pm 36.9$ | $2400 \pm 8.65$ |
| NF-MCMC | $-26.9 \pm 9.62$ | $548 \pm 325$ | $549 \pm 325$ | $2000 \pm 15.6$ |
| FAB | $-50.4 \pm 0.14$ | $0.14 \pm 0.42$ | $1.78 \pm 0.42$ | $3880 \pm 7.19$ |
| AT-SMC | $-63.5 \pm 9.91$ | $1.61 \pm 2.33$ | $18.4 \pm 2.35$ | $4.13 \pm 0.30$ |

Table 9: Diagnostics for the field system with $K = 10^4$. $\mathbb{E}_{[\phi_1]_{\#}p_0} \log \pi$ is the Monte Carlo approximation of the log-target density using the learned flow to generate samples; KSD U-stat and V-stat are the Kernel Stein Discrepancy U- and V-statistics between the target and samples generated from the learned flow. Results are averaged and empirical 95% confidence intervals over 10 independent runs.

i.e. $X \sim \mathcal{N}(\mu_0 1_d, \Sigma_0)$ for $1_d = [1, \ldots, 1]^T \in \mathbb{R}^d$ with dimension $d = 1600$. The chosen parameter values are $\sigma^2 = 1.91$, $\beta = 1/33$, and $\mu_0 = \log(126) - \sigma^2/2$, corresponding to the values estimated in [53]. The number of points in each grid cell $Y = \{Y_m\}_{m \in M} \in \mathbb{N}^{40 \times 40}$ are modelled as conditionally independent and Poisson distributed with means $a\Lambda_m$,

$$\mathcal{L}(Y|X) = \prod_{m \in [1:30]^2} \exp(x_m y_m - a \exp(x_m)), \tag{44}$$

where $a = 1/40^2$ represents the area of each grid cell. For this example, all methods use $N = 128$ parallel chains for training and 1024 hidden dimensions for all neural networks. Methods with a MALA kernel use a step size of 0.01, and methods with splines use 8 coupling layers with 8 bins and range limited to $[-10, 10]$. Results for $K = 10^4$ learning iterations are presented in Table 10. In the main text, we present results for $k_Q = 10^3$ for MFM. We were unable to run NF-MCMC and FAB for $K = 10^4$ iterations because of memory issues; instead, we present results for $K = 10^3$ iterations only for the models using discrete normalizing flows.

| $K = 10^4$ | $\mathbb{E}_{[\phi_1]_{\#}p_0} \log \pi$ | KSD U-stat. | KSD V-stat. | seconds |
|---|---|---|---|---|
| MFM $k_Q = K$ | $-1960 \pm 10.8$ | 1.13e-1$\pm$5.18e-2 | $28.1 \pm 0.246$ | $117 \pm 4.19$ |
| MFM $k_Q = 10^3$ | $-1960 \pm 10.8$ | 1.14e-1$\pm$5.13e-2 | $28.1 \pm 0.241$ | $1690 \pm 870$ |
| – w/ HTE | $-1960 \pm 12.2$ | 1.12e-1$\pm$4.93e-2 | $28.1 \pm 0.236$ | $143 \pm 14.5$ |
| MFM $k_Q = 10^2$ | $-1960 \pm 12.6$ | 1.15e-1$\pm$5.12e-2 | $28.0 \pm 0.265$ | $17300 \pm 9970$ |
| – w/ HTE | $-1960 \pm 11.1$ | 1.15e-1$\pm$5.17e-2 | $28.1 \pm 0.239$ | $394 \pm 157$ |
| DDS | $-1850 \pm 8.59$ | 7.59e-2$\pm$2.24e-2 | $24.7 \pm 0.08$ | $3260 \pm 8.41$ |
| NF-MCMC | $-1410 \pm 53.8$ | $11.8 \pm 7.55$ | $89.0 \pm 238$ | $215 \pm 46.4$ |
| FAB | $-3070 \pm 80.7$ | 1.55e-1$\pm$6.12e-2 | $52.3 \pm 2.02$ | $1040 \pm 2.78$ |
| AT-SMC | $-1910 \pm 4.21$ | 1.39e-2$\pm$1.14e-2 | $25.0 \pm 0.12$ | $6.11 \pm 0.44$ |

Table 10: Log Gaussian Cox point process diagnostics for $K = 10^4$ where $\mathbb{E}_{[\phi_1]_{\#}p_0} \log \pi$ is the Monte Carlo approximation of the log-target density using the learned flow to generate samples; KSD U-stat and V-stat are the Kernelized Stein discrepancy's U- and V-statistics between the target and samples generated from the learned flow. Results are averaged and empirical 95% confidence intervals over 10 independent runs.

