# OpenReview forum: "Markovian Flow Matching: Accelerating MCMC with Continuous Normalizing Flows"
_NeurIPS.cc/2024/Conference — NeurIPS 2024 poster_

### Official Review · Reviewer_Sj6c · 2024-07-02

**Soundness:** 3
**Presentation:** 3
**Contribution:** 2
**Rating:** 6
**Confidence:** 3

**Summary:**

The proposed method falls in the wider category of models that integrate MCMC methods with normalizing flows. Specifically, the method expands on the neutra-MCMC model, where the key idea is to run MCMC in a simpler reference space rather than in the (more complicated) target one. The main difference is that previous work uses discrete normalizing flows while the proposed approaches employs the Flow Matching framework and hence relies on a continuous velocity field, which is provided with a continuous Normalizing Flow.

**Strengths:**

- the authors propose an interesting way to use the Flow Matching framework in the context of Flow-enhanced MCMC samplers
- I also found interesting the use of a sample-based annealing scheme (which is however not novel). The authors claim it works particularly well for multi-modal distributions, which is definitely a very relevant setting

**Weaknesses:**

- the novelty aspect of the proposed method seem limited to extend current formalism of MCMC - Normalizing Flows (NFs) methods (specifically neutra-MCMC) to the specific architecture of Continuous Normalizing Flows (CNF). However, I don't see a priori why the (continuous) CNF should be preferable discrete NFs (which are even more expressive than CNF [1]) if not because the flow matching objective requires access to the velocity field provided by CNF

[1] Perugachi-Diaz et al., Invertible DenseNets with Concatenated LipSwish, NeurIPS 2021

**Questions:**

1. Some work analyzes the performance of methods combining MCMC with NFs methods in terms of (i) how easy is it to sample from multi modal distributions and (ii) scalability with dimensions [1]. They found that neutra-MCMC (similarly to the proposed method) is efficient in multi-modal settings but struggles more in high dimensions, compared to methods that use Flows as proposal for MH. Did you observe a consistent behaviour with their analysis? For instance I would have expected NF-MCMC to perform better in the Log-Gaussian Cox experiment. Or did the proposed method outperform "flow-MCMC" methods also when inflating the dimensions even more? (e.g. in the experiments in Section 5.3 and 5.4)
2. The use of CNF are motivated by the availability of the velocity field, which is required by the Flow Matching objective. However, other neutra-MCMC methods work very well also with discrete NFs. CMF are definitely more expressive than standard discrete flows that are commonly used in the literature (mainly real NVP), but one could also use more powerful discrete NFs like [2]. I was wondering whether the authors could discuss the benefits of the proposed approach compared to neutra-MCMC with discrete NFs.
3. The proposed method uses a hybrid between local and global updates, similarly to [3] (which is cited in the paper). The main difference it the use of a CNF (for the Flow Matching objective) instead of NFs. So I think it would be very relevant to compare the proposed method against [3]. Also the authors of [3] use very naive layers (real NVP, which consist of basic shift and scale transforms), while more flexible layers like [2] might exclude that the difference in performance is to be connected to the expressiveness of the flows. Similarly, the NF-MCMC uses very simple NFs layers (always real NVP).
4. Even though the Flow Matching objective could allow to learn complicated distributions, in the end the authors use a very simple velocity field to transform the conditional distribution. Doesn't this limit the expressiveness of the learnt distribution?
5. In line 218 do the authors mean FAB instead of DDS?


[1] Grenioux et al, On Sampling with Approximate Transport Maps, ICML 2023

[2] Perugachi-Diaz et al., Invertible DenseNets with Concatenated LipSwish, NeurIPS 2021

[3] Samsonov, Local-Global MCMC kernels: the best of both worlds, NeurIPS 2022

**Limitations:**

Limitations are clearly stated.

---

> ### Author Rebuttal · Authors · 2024-08-01
>
> We would like to thank the reviewer for their positive comments, and for their detailed and constructive feedback. We provide a point by point response to their review below.
>
> We hope that we have been able to address the reviewers' questions, and that they will consider increasing their score based on our response and changes to the paper. We look forward to engaging with them during the discussion period, and welcome any further questions.
>
> **Weaknesses**
> - **The novelty aspect of the proposed method seem limited...** - We are grateful to the reviewer for pointing out [1], although we would contend that [1] does not show discrete NFs are more expressive than CNFs. While the residual LipSwish architecture in [1] is more expressive than the standard coupling flow used by discrete NFs, it is still constrained to a specific architecture which cannot be freely adapted to a specific model of interest. A significant strength of our method is that it can use any architecture and, in particular, one which can be tailored to the target distribution at hand. This being said, we agree that an interesting avenue for future work would be a more detailed evaluation of how CNFs perform in comparison to the most expressive discrete NFs (e.g., LipSwish).
>
> **Questions**
>
> **1.** Thanks for highlighting the recent benchmarking paper [2]. This paper did indeed influence our choice to compare our method to NF-MCMC rather than a neutra-MCMC method (e.g., transport elliptical slice sampling [3]). In particular, as the reviewer notes, [2] shows that NF-MCMC generally scales better with dimension than neutra-MCMC. Given this, we thought that NF-MCMC would be a fairer comparison in high-dimensional experiments. Interestingly, in our experiments, we found that NF-MCMC scaled worse than FAB, as well as worse than MFM and DDS. We suspect that MFM (our method) outperforms NF-MCMC in these high-dimensional settings as MFM is not a "pure" neutra-MCMC method. In particular, we combine a Markov transition kernel which targets the pullback of the target measure under our CNF (similar to other neutra-MCMC methods) with a local transition kernel which targets the target measure directly (similar to, e.g., [4]) and an adaptive annealing strategy.
>
> **2.** As noted in our response above (see "weaknesses"), one strength of our method (and, more generally, of CNFs) is that the neural network architecture is entirely unconstrained. In particular, there is the possibility of tailoring the architecture to the model (i.e., the target distribution) at hand, since we are not constrained to a specific Jacobian. This is not the case for discrete NFs,  even more expressive discrete NFs with more exotic architectures such as the one proposed in [1].
>
> **3.** We agree with the reviewer that [5], which combines a flow-informed global transition kernel (i-SIR) and a local transition kernel, is a very relevant reference. It is worth noting that the algorithmic template in [5] is very similar to the one in [4], which was (to our knowledge) the first paper to combine a flow-informed global transition kernel (IMH) with a local transition kernel (MALA). Interestingly, among the stated reasons that [4] focus on i-SIR for their global update as opposed to IMH (as proposed in [5]) is that "theoretical guarantees can be obtained for i-SIR whereas IMH is more difficult to analyze" and "IMH and i-SIR (as a multiple-try MCMC) are expected to have similar performances for comparable computational budget" [4, Section 4]. Given this, we felt that including the method proposed in [5] (NF-MCMC) as a comparator in our numerical experiments was sufficient. With regards to the neural network architecture used for the methods in [4, 5], we used the simple layers (realNVP) implemented in their paper (and in the accompanying software). We believe that this is a fair comparison, given that we also use a relatively simple neural architecture for our CNF. While beyond the scope of this paper, we agree with the reviewer that an interesting avenue for future work would be to investigate whether using more flexible layers for the discrete NFs used in (e.g.,) [4,5] would enhance the performance of these methods.
>
> **4.** The reviewer is correct that, throughout our experiments, we use a relatively simple parametrisation of the vector field. While this does limit the expressiveness of the learned distribution, we found in our experiments that it was sufficient to achieve comparable performance to competing methods, often for a fraction of the computational cost. The design of novel CNF architectures, tailored to specific models, is a non-trivial question, beyond the scope of this paper. However, this would be a very interesting direction for future work. Importantly, there are *no* restrictions on the neural network architecture utilised by our method, so there is significant scope for exploring other architectures which may further improve performance. To illustrate this point, in our global response (see attached PDF), we provide additional results when using a neural network which includes an additional time-dependent weighting (similar to DDS). These results show a significant improvement over the results reported in the original submission, and illustrate the scope for additional improvement based on further refinements of the neural network used by the CNF.
>
> **5**. Thanks to the reviewer for pointing this out, we did indeed mean FAB in L281. This has now been corrected!
>
> **References**
>
> [1] Perugachi-Diaz et al., 2023. Invertible DenseNets with Concatenated LipSwish. NeurIPS 2021
>
> [2] Grenioux et al., 2023. On Sampling with Approximate Transport Maps. ICML 2023.
>
> [3] Cabezas et al., 2023. Transport Elliptical Slice Sampling. AISTATS 2023.
>
> [4] Gabrie et al., 2022. Adaptive Monte Carlo Augmented with Normalizing Flows. PNAS.
>
> [5] Samsonov et al., 2022. Local-Global MCMC Kernels: The Best of Both Worlds. NeurIPS 2022.

---

> > ### Comment · Reviewer_Sj6c · 2024-08-14
> >
> > I would like to thank the authors for their clarifications and for providing additional evaluation of their method. I have adjusted my score to reflect these changes.

---

### Official Review · Reviewer_HJhZ · 2024-07-13

**Soundness:** 3
**Presentation:** 3
**Contribution:** 2
**Rating:** 5
**Confidence:** 4

**Summary:**

A method called Markovian flow matching (MFM) for training neural ODEs (continuous normalizing flows) by flow matching to sample from distributions given as unnormalized density functions is proposed. The method obtains samples on which to perform FM training by an MCMC procedure that alternates two kinds of kernels: Metropolis-adjusted Langevin in the target space (as has been done in past work) and Gaussian proposals in the latent space, i.e., the pullback of the target by the learned ODE. An annealing scheme is additionally used. MFM is evaluated on toy low-dimensional tasks, a synthetic density from statistical physics, and a high-dimensional data-derived density (log-Gaussian Cox process).

**Strengths:**

- The idea to do MCMC in the latent space of a CNF by pullback along the flow map is original, as far as I know.
  - There are two general families methods for training continuous-time models to sample Boltzmann densities: using maximum likelihood on (possibly reweighted) approximate samples (e.g., FAB and the proposed MFM) and using distribution-matching objectives (e.g., DDS, PIS), as well as hybrid approaches.
  - It's interesting to see progress on making methods of the first kind efficient, and to see them combined with CNFs. In the past, e.g., in [Tong et al. "Improving and generalizing flow-based generative models", D.2], FM has been used with MCMC, but only in the target space.
- No serious problems with the writing, including the global structure of the paper and the exposition of preliminaries.

**Weaknesses:**

- Some math bugs (imprecise exposition):
  - L72: $v_t$ being **any** time-dependent vector field does not imply existence of a diffeomorphic flow map (even if it is continuous in $t$, if that is how we interpret "runs continuously in the unit interval"). One also needs integrability conditions.
  - L75: $p_0$ has to be strictly positive for the path to take values in $\mathbb{R}^+$ and for the discussion the discussion that follows to make sense.
- Comparison with SDE models: Approximate (MCMC) samples can also be used to train diffusion models (neural SDEs):
  - First, this could be done either by minimising a variational bound on the log-likelihood, similar to what is done here for CNFs -- is it possible to do such a comparison? At present the SDE baseline (DDS) is not using MCMC, only "on-policy" forward exploration.
  - Second, methods such as that in [Sendera et al. "Improved off-policy training of diffusion samplers"] use MCMC on the target density to obtain samples for training with a distribution-matching, not maximum-likelihood, objective.
- The main weakness is that the experiments are not very comprehensive or convincing:
  - The 2D GMMs are toy problems; all methods, including DDS, should find the modes when appropriately tuned on such problems, and so it is hard to draw any conclusions from the results.
  - On the field system, MFM, although much faster than FAB, is performing **much** worse in the metrics used.
  - On LGCP:
    - The use of this task for benchmarking samplers of Boltzmann densities is questionable in the first place, since there is no known ground truth (even the exact normalising constant is diagreed upon) -- see, for example, the aforementioned [Sendera et al., B.1] for a discussion of inconsistent or irreproducible evaluation in past work on this dataset.
    - Why use the nonstandard metric (KSD)? Most past work has used estimates of the log-partition function, which here is reported in the appendix.
    - On that more standard metric, MFM is significantly underperforming (Table 8).
- On both the field system and LGCP, how are we to know if the proposed method is useful on this problem without a point of comparison? I would strongly suggest a long-run HMC, MALA, and SMC (for example, run for the same wall time as the shortest and longest of the ML methods evaluated) as baselines.
- I would also strongly suggest evaluations on a more comprehensive set of benchmarks that are common in past work on samplers of Boltzmann densities (as in the FAB and DDS papers, among others).

**Questions:**

Minor:
- L25: You probably want to specify "a new sample $y$" and in the subsequent lines clarify that the "mild conditions" are also needed for the target to be the *only* stationary distribution.
- L77: Dash between "change" and "of" instead of hyphen.
- L86: Wrong placement of parentheses.
- L246: I do not believe [78] introduced this parametrisation -- it was in [80] (and possibly in earlier work as well).
- I don't understand equation (18), which involves $x_{65}$ in the first sum although the dimensionality is 64. Is there is mistake, or should the indices be interpreted modulo $d$?
- L308: Typo in "comparable".

**Limitations:**

Yes

---

> ### Author Rebuttal · Authors · 2024-08-01
>
> Many thanks to the reviewer for their thorough engagement with our work and for their constructive feedback. We provide a detailed point-by-point response below.
>
> We hope that we have been able to address the reviewers' questions, and that they will consider increasing their score based on our response and changes to the paper. We look forward to engaging with them during the discussion period, and welcome any further questions.
>
> **Weaknesses**
> - **Some math bugs...**  We agree we should have been more precise here. We have now amended L72 and L75 to include the required conditions, as suggested by the reviewer.
> - **Comparison with SDE models...** Thanks for pointing this out. With regards to *minimising a variational bound on the log-likelihood using (MCMC) samples,* the problem here is that we need to solve the ODE numerically both for generating MCMC samples and evaluating the gradient of the log-likelihood. This means such a method would be very inefficient since the ODE solver is the most expensive part. This is one of the motivations for using flow matching; we don't need to solve the ODE, just evaluate the vector field. With regards to *off-policy training*, we were unaware of the literature the reviewer cited [1]. We agree that it would be nice to (empirically) compare our method with an off-policy method, although this was not possible during the rebuttal period. We have updated the related work section to include this reference and will aim to include this as a benchmark in the revised version of the paper.
> - **The main weakness is that the experiments are not very comprehensive...**
> 	- With regards to the two-dimensional experiments, similar experiments are a standard benchmark in similar papers, including recent works such as [2,3]. In the 16-mode Gaussian mixture experiment (Section 5.2), we were unable to find a hyperparameter configuration in which DDS captured all of the modes.
> 	- With regards to the field system, while FAB does outperform our method with respect to KSD, FAB does *not* in fact capture both of the modes (see Figure 3).  Indeed, of the considered methods, only our approach (MFM) captures both of the modes in this example. In addition, the runtime for MFM is *significantly* lower than that of FAB (see the discussion in Section 5.3).
> 	- With regards to LGCP, we were unaware of the discussion regarding the suitability of this task as a benchmark for sampling methods [1, Section B.1], and are grateful to the reviewer for bringing this to our attention. It is worth noting that this benchmark has been used widely in several other recent papers in this field, hence its inclusion in our paper [e.g., 4]. In terms of our use of the KSD (and the MMD), we do not think these are particularly non-standard metrics. Their use is largely justified by [5] and subsequent work; see also recent works including [6,7,8] or [9] where the KSD or MMD are used as metrics for assessing sample quality. Regardless, in our updated results, included in the PDF in the global response, our method in fact performs similarly to SMC when measured in terms of $\\smash{\\mathbb{E}_{[\\phi_1^{\theta}]{\\#}p_0} [\log \\pi]}$. These results are obtained by including a time-dependent weight in our neural network for the vector field (as in DDS) and illustrate that there is further room for improvement in our method by exploring other neural network architectures. This is a key strength of our approach: it allows the flexibility to use any vector field architecture (since the architecture is entirely unconstrained for CNFs), which is not the case for DDS. We leave a more detailed investigation of this to future work.
> - **On both the field system..., how are we to know if the proposed method is useful...without a point of comparison?** We agree that the inclusion of an additional 'ground truth' benchmark would be useful here. We have now generated additional results for these experiments using (adaptively tempered) SMC. These results are included in the PDF in our global response. We will include these results in the revised version of the paper, with the caveat that there is some debate on the quality of such a benchmark in such a high-dimensional problem.
> - **I would... suggest evaluations on a more comprehensive set of benchmarks...**  We have now also added results for the 'many wells' experiments in [2]. With the inclusion of the experiment, we would argue that our numerical experiments cover a significant number of the experiments considered in other recent papers of a similar flavour (e.g., FAB, NF-MCMC, DDS). In particular, the 16-mode example and the 'many wells' experiment are considered in [2], the field system example in [10], and the LGCP in [5].
>
> **Questions**
> - **Minor**. Thanks for pointing these out. We have amended the manuscript as per all of your suggestions. Regarding (18), your second suggestion (i.e., that the indices should be interpreted modulo $d$ is correct). We have added a short remark to clarify this.
>
> **References**
>
> [1] Sendera et al., 2024. Improved Off-Policy Training of Diffusion Samplers. arXiv.
>
> [2] Midgley et al., 2023. Flow Annealed Importance Sampling Bootstrap. ICLR 2023.
>
> [3] Arkhound-Sadegh et al., 2024. Iterated denoising energy matching for sampling from Boltzmann densities. ICML 2024.
>
> [4] Vargas et al., 2023. Denoising Diffusion Samplers. ICLR 2023.
>
> [5] Gorham et al., 2017. Measuring Sample Quality with Kernels. ICML 2017.
>
> [6] Nemeth et al., 2021. Stochastic Gradient Markov Chain Monte Carlo. JASA.
>
> [7] Chehab et al., 2024. A Practical Diffusion Path for Sampling. Chehab et al., 2024. ICML 2024: SPIGM Workshop.
>
> [8] Maurais et al., 2024. Sampling in Unit Time with Kernel Fisher–Rao Flow. ICML 2024.
>
> [9] Blessing et al., 2024. Beyond ELBOs: A Large Scale Evaluation of Variational Methods for Sampling. arXiv.
>
> [10] Gabrie et al., 2022. Adaptive Monte Carlo Augmented with Normalizing Flows. PNAS.

---

> > ### Comment · Reviewer_HJhZ · 2024-08-12
> >
> > Thank you for the clarifications.

---

### Official Review · Reviewer_eFeW · 2024-07-15

**Soundness:** 4
**Presentation:** 4
**Contribution:** 3
**Rating:** 7
**Confidence:** 4

**Summary:**

The authors propose a novel method for incorporating flow matching with MCMC; entailing constructing a Markov kernel as a mixture of regular MCMC step and flow step (from data space (\pi_1) to prior (\pi_0, typically Gaussian) and back with some added noise. The flow guarantees a likelihood and can hence be used for accept/ reject corrections.

In this sampling setting, one does not have access to samples from the data and hence the flow is trained from MCMC samples jointly within sampling.

**Strengths:**

I think the authors did a great job on this paper:
- The method is simple and well explained
- Experiments are thorough with appropriate baselines including recent DDS, good performance and code is available
- The authors clearly articulate limitations, which I appreciate
- Prior and related work is discussed and not hidden

**Weaknesses:**

I worry how long it takes the flow network to converge. Simulating the flow back and forward to generate samples at regular frequency during training could be quite slow if involving a large number of network evaluations for the flow. And if the MCMC locally does not mix then the flow will not see many samples. I understand tempering may help resolve this but at what cost?

Similarly, I imagine the samples from the flow could be quite far from the target distribution if the flow was not pre-trained or initialized to be close to some nearby distribution.

The idea itself is quite straightforward. This is not necessarily a weakness in itself, I am surprised something similar has not been considered which am unaware of. I understand there are many similar schemes that are cited using non-local Markov updates with normalizing flows (discrete), just not using flow matching (e.g. https://arxiv.org/abs/2105.12603) and other works from the same authors. I appreciate the authors have discussed and cited these. This however limits the novelty of the methodological contribution slightly in my opinion.


Minor presentation:
- I struggle to understand from Figure  what the ground truth samples should look like. I assume like the samples from the author's method. It would help readers to show the density to know what the ground truth should be, similar to the other experiments.
- Similarly Table 2, show in bold the top performing items to help readers

Some other relevant works using diffusion / flow for sampling:
- Target score matching, Bortoli et al 2024 https://arxiv.org/abs/2402.08667
- Iterated Denoising Energy Matching for Sampling from Boltzmann Densities, Akhound-Sadegh et al 2024
and although cited the work of Phillips et al 2024 (already cited) could be used as a baseline given code is available (also in jax)
- CRAFT and AFT  (Annealed flow transport Monte Carlo, 2021) may also be good baselines to include, code is available in jax for those too.

**Questions:**

My understanding is that the flow update pushes samples into a simpler prior / latent type space then back to the data space. Is it possible to include Markov transitions in the prior space i.e. in the simpler (Gaussian) distribution, which are easier and encourage non local updates (in data space)? The addition of Gaussian noise is a bit like this I suppose but has anything else been considered?


Does this actually require flow matching? I feel any diffusion model would also have be usable here, it can be trained in the same way and one can use the probability flow ODE to have tractable likelihoods and if needed similar to flow matching. There are many similarities between the two models, especially for Gaussian marginal, but diffusions seem to be better performing generative models still, see e.g. Lipman 2022 (cited) vs Karras 2022 (https://arxiv.org/abs/2206.00364) in terms of generative performance. If just using Langevin dynamics without accept/reject correction then the added stochasticity of diffusion samplers may also help, plus the benefit of Langevin dynamics corrector steps along the diffusion paths could possibly be used (Song 2021).

**Limitations:**

See weaknesses.

---

> ### Author Rebuttal · Authors · 2024-08-01
>
> We thank the reviewer for their positive feedback and constructive comments. We provide a detailed point by point response to their review below.
>
> **Weaknesses**
> - **I worry how long it takes the flow network...** The reviewer is correct that flow-informed MCMC samples are relatively expensive to compute, due to the need to simulate the CNF. However, the rest of the algorithm is very inexpensive. Moreover, our empirical results illustrate that we do not require a very large number of flow-informed transition steps for the flow network to converge to a good approximation of the target distribution, even when the flow network is not initialised close to the target distribution. It is worth noting that, given particular computational constraints, one can balance the (more expensive) flow-informed global updates with the (much less expensive) local updates by varying the value of $k_Q$ (see Appendix C for some indicative results).
> - **Similarly, I imagine samples from the flow could be quite far from the target...if the flow was not pre-trained...** The reviewer raises an interesting point with regards to pre-training of the flow, or initialisation of the flow network. In cases where one has some a priori information about the target distribution (e.g., the location of the modes), our algorithm could instead use a reference distribution which utilises this information (e.g., a mixture of Gaussians, centred at these modes). This is possible using the general formulation of flow matching given in, e.g., [1], which allows for the arbitrary specification of reference and target distributions and may further improve the convergence rate of the flow network in our algorithm. However, in all of our experiments, where a standard Gaussian was used as the reference density, (samples output by) the flow network converged to the target within a reasonable number of iterations.
> - **Minor presentation...** Thanks for pointing out these issues. We have now included the ground truth samples in this figure and ensured that the best-performing method is shown in bold in Table 2 (and all other tables).
> - **Some other relevant works...** Thanks for pointing out these papers. We have now added [2] and [3] to our discussion of related work. Regarding baselines for our numerical experiments, we would argue that the baselines we have already included cover the most relevant methods in the literature. Nonetheless, we agree that including results for the methods in [4, 5, 6] would also be useful. Given the time constraints, we have not been able to re-run all of our numerical experiments for these methods during the rebuttal period. However, we will look to include these experiments in the revised version of our paper. During the rebuttal period, we *have* obtained results for all of our numerical experiments for an adaptively tempered SMC scheme, which are included in the PDF in our global response to all reviewers.
>
> **Questions**
> - **My understanding is the flow pushes samples into a simpler...** The reviewer is correct that other Markov transitions are also possible in the latent space, and we have now added a remark in Section 3 to ensure this is clear. For example, there is also the option to do independent MH (instead of RWMH). However, we found that the added noise of the RWMH transition kernel helped with exploration of the state space, and avoiding mode collapse. Although, due to space constraints, we are unable to include these results in the PDF attached to our global response to all reviewers, we will include this comparison in an appendix in the revised version of the paper. Other options that use local gradient information are also possible, but these would require even more evaluations of the pullback target density (and its gradient). This requires numerically solving the ODE, which is by far the most expensive part of the algorithm.
> - **Does this actually require flow matching?** The reviewer is correct that one could use a diffusion model within our algorithm, using the probability flow ODE to evaluate the likelihood of the samples under the model. On the one hand, this would allow for other sampling schemes (e.g., stochastic samplers, predictor-corrector schemes, etc.) as the reviewer suggests. It would certainly be interesting to investigate further whether the use of such samplers has any benefit in terms of algorithmic performance, although we feel that this is beyond the scope of this work. On the other hand, the use of (conditional) flow matching arguably allows for more flexibility in terms of the specification of the probability path, as well as arbitrary source distributions [1].
>
> Many thanks again to the reviewer for their useful feedback. We hope that we have been able to fully answer their questions and that our responses will increase their confidence in this paper.
>
> **References**
>
> [1] Tong et al., 2024. Improving and generalizing flow-based generative models with minibatch optimal transport. TMLR.
>
> [2] Bortoli et al, 2024. Target Score Matching. arXiv.
>
> [3] Arkhound-Sadegh et al., 2024. Iterated denoising energy matching for sampling from Boltzmann densities. ICML 2024.
>
> [4] Phillips et al, 2024. Particle Denoising Diffusion Sampler. ICML 2024.
>
> [5] Arbel et al., 2021. Annealed Flow transport Monte Carlo. ICML 2021.
>
> [6] Matthews et al., 2022. Continual Repeated Annealed Flow Transport Monte Carlo. ICML 2022.

---

### Official Review · Reviewer_YCu1 · 2024-07-17

**Soundness:** 3
**Presentation:** 4
**Contribution:** 3
**Rating:** 7
**Confidence:** 3

**Summary:**

This paper aims to use continuous normalizing flows (CNFs) to define the proposal distribution in a MCMC framework. While the use of flow models for MCMC proposals is not entirely new, this paper introduces an interesting training procedure that iteratively updates the learned CNF model while performing MCMC. Existing works either (i) make use of an importance weighted training objective, or (ii) first perform long-run MCMC to obtain high-quality samples. Instead, this work makes use of the Flow Matching training objective to fit to its current MCMC samples, where samples are obtained from a MCMC involving the CNF itself.

**Strengths:**

- Overall, I feel this paper is a combination of good solid ideas that individually are not entirely new, but packaged in a way that makes sense.
- The writing is also clear and to the point.
- Experiments seem to be carried out on some standard benchmark potential functions.
- The paper repeatedly mentions that the proposed method is favorable to competing neural MCMC approaches based on wallclock time, being significantly faster in some cases.

**Weaknesses:**

- I feel the set of experiments is "too standard". Given that Bayesian inference is such a long field, it would be make a paper much stronger if it can show meaningful improvement on real unsolved problems where exploration is key.
- There is a need to add ablation experiments that provide better intuition regarding what benefit the flow-informed Markov transition is doing.

**Questions:**

Generally, while the proposed MCMC algorithm as a whole makes sense, it is composed of multiple components. In order to better understand each proposed component, it would be good to have ablation experiments analyzing the behavior of each component.

- Given that there is a series of local MCMC updates, it would be good to compare against MCMC with just the Q transition kernel. This can help answer what exactly is the change in improvement from including a flow-informed transition kernel.

- I think what would be really interesting is a plot of the acceptance rate between the P (flow-informed) and Q (local) kernels. I would imagine that at the beginning P has poor acceptance rates because the CNF is poorly trained, while after the MCMC chain is run for long enough times, its acceptance rates are much higher. Meanwhile, the acceptance rate of Q should be constant but perhaps lower than the acceptance rate of P? Regardless, it would be interesting to see such a plot to better understand the behavior of the proposed algorithm.

- If you turn off the annealing, how much worse does the approach perform?

### Minor comments
- The likelihood notation L(D|x)  when discussing annealing (Eq 16) is not used anywhere else (and also not used in the experiments since they directly define pi(x). I felt this to be confusing because as it is currently written, it may seem that the annealing trick is restricted to only Bayesian inference settings. I would imagine the annealing simply takes L(x) = pi(x) / pi_0(x) for some pi_0(x). It might be good to clarify this?
- There are 3 instances of "NeutraMCMC" which should probably be "Neural MCMC" ?

**Limitations:**

The paper is written clearly with objective analyses, with no excessive beautification of the proposed method. It's nice.

---

> ### Author Rebuttal · Authors · 2024-08-01
>
> We are grateful to the reviewer for their positive remarks and their constructive feedback. We provide a detailed point by point response to their review below.
>
> **Weaknesses**
> - **I feel the set of experiments is too standard...** Thanks for raising this comment. While we agree with the reviewer that the inclusion of real-world unsolved inference problems would further strengthen this work, we believe that this is beyond the scope of the paper. Indeed, we would argue that the set of numerical experiments presented are very comparable to other recent works in the literature [1,2,3]. Not only this, but they demonstrate the robust performance of our method in a range of challenging examples (e.g., multimodal targets, high-dimensional targets, etc.).
> - **There is a need to add ablation experiments...** We agree with the reviewer that some additional ablation experiments would help to elucidate the contribution of different components of our algorithm. In our latest revision of the paper, we now include several such experiments. We provide full details in our responses to the points raised in the 'questions' section of the review (see below).
>
> **Questions**
> - **It would be good to compare against MCMC with just the Q transition kernel...**  This comparison is actually already present for all of our existing numerical experiments. In particular, results for running the algorithm with just the Q transition kernel correspond to the results with k_Q=K (see, e.g., Table 3 - 8 in Appendix C). We have now added an additional remark at the start of the numerical experiments section to clarify this, and moved these results from the appendix to the main paper.
> - **I think what would be interesting is a plot of the acceptance rate...** We are grateful to the reviewer for this suggestion, and agree that it would be interesting. The acceptance rate of Q is targeting 1 since we estimate expectations, e.g. in (14), using the current ensemble of particles, rather than a single long chain (see L244). The acceptance rate of P certainly increases as the algorithm runs; in particular, it often jumps to high values, before stabilising to a lower value. We can include plots to illustrate this behaviour in an appendix in the revised version of the paper.
> - **If you turn off the annealing, how much worse does the approach perform?** Thanks to the reviewer for raising this interesting question. The relative performance of the algorithm with or without annealing is dependent on (i) the task at hand and (ii) the initialisation of the particles. In particular, if the initialisation of the particles doesn’t cover one or more of the target modes, then these modes are much less likely to be explored without annealing. In general, without the annealing strategy, one shouldn't expect the algorithm to find states in modes distinct from initialisation. See also the related discussion in [3, Section IV.C]. On the other hand, when at least one particle is initialised in each of the the metastable basins (i.e., the modes) of the target distribution, annealing is much less important. To confirm this behaviour, we have run our algorithm with and without annealing for one of the multimodal two-dimensional experiments described in Section 5. These results, which indeed confirm the behaviour discussed above, are included in an appendix in the revised version of our paper.
>
> - **The likelihood notation...** The use of the likelihood notation in (16) is indeed not used elsewhere, and we agree with the reviewer that this notation may suggest the use of annealing is only restricted to the setting of Bayesian inference. As the reviewer correctly points out, this is not the case. We have now rewritten this section in to cover the general setting.
> - **There are 3 instances of neutraMCMC...** The term "NeutraMCMC" is actually correct here. This terminology originates in [4]; see also [5, Section 2.2].
>
> Many thanks again to the reviewer for their useful feedback. We hope that we have been able to fully answer their questions and that our responses will increase their confidence in this paper.
>
> **References**
>
> [1] Midgley et al., 2023. Flow Annealed Importance Sampling Bootstrap. ICLR 2023.
>
> [2] Vargas et al., 2023. Denoising Diffusion Samplers. ICLR 2023.
>
> [3] Gabrie et al., 2022. Adaptive Monte Carlo Augmented with Normalizing Flows. PNAS.
>
> [4] Hoffman et al., 2019. Neutra-lizing Bad Geometry in Hamiltonian Monte Carlo using Neural Tranport. AABI 2018.
>
> [5] Grenioux et al., 2023. On Sampling with Approximate Transport Maps. ICML 2023.

---

### Author Rebuttal · Authors · 2024-08-01

**Summary**

Many thanks to all of the reviewers for their positive feedback about the paper, as well as their detailed and constructive comments. We provided a detailed point-by-point response to each of the reviewers' specific comments in the individual responses below.

**Additional Results**

We attach to this global response a set of additional numerical results, which we have generated based on the feedback of one or more of the reviewers. In particular, the attached PDF contains:

- New results for all of the numerical experiments using a slightly different parameterisation of the vector field, namely, $\text{NN}^*(t; \theta_3) v_t^\theta(x) = \text{NN}(x, t; \theta_1) + \text{NN}(t; \theta_2) \times \nabla \log \pi(x)$, where the neural networks are standard MLPs with 2 hidden layers, using a Fourier feature augmentation for $t$, and $\text{NN}^*$ outputs a real value that reweights the vector field output using the time component. These new results show a significant improvement over the results in the original submission, and illustrate the scope for additional improvements in the performance of MFM (our algorithm) based on further refinements to the design of the neural network architecture used to parameterise the vector field. These results appear in Tables 1,2,4,5 in the PDF.

- New results for an additional benchmarking experiment, namely, the "Many Well" experiment considered in [1,2,3]. These appear in Table 3 in the PDF.

- New results for all of the numerical experiments for an adaptively tempered sequential Monte Carlo (SMC) algorithm (AT-SMC). These appear in the bottom row of all of the tables in the PDF.

- Results for all of the numerical experiments when running MFM with just the 𝑄 transition kernel. These results appear in the row labelled "MFM $k_Q = K$" in the PDF.


**References**

[1] Noé et al., 2019. Boltzmann generators: Sampling Equilibrium States of Many-Body Systems with Deep Learning. Science.

[2] Wu et al., 2020. Stochastic Normalizing Flows. NeurIPS 2020.

[3] Midgley et al., 2023. Flow Annealed Importance Sampling Bootstrap. ICLR 2023.

---

### Decision · Program_Chairs · 2024-09-25

**Decision:**

Accept (poster)

**Comment:**

This paper suggests using flow matching trained CNFs to build MCMC proposals. The main novelty seems to be the incorporation of the flow matching training of the CNF simultaneously with the MCMC process, using generated MCMC samples.

Reviewers mostly agreed this paper has merit and offers an interesting contribution; has clear exposition; presents a reasonable set of experiments supporting the paper main claims (i.e., method is comparable-to-favorable and often much faster than alternative MCMC sampling baselines); limitations are clearly presented; prior work is well explained and positioned.

Regarding limitations of the paper: Reviewers asked about missing ablations and some limitation in the extent of experiments demonstrated, which the authors addressed, at least partially (which is reasonable within the rebuttal time frame) in the rebuttal, including a new set of additional numerical results; the convergence time of flows due to the need to sample from the model during training, which the authors addressed by confirming this is indeed an issue, yet still worthwhile overall as number of iteration is not large; missing some relevant work and discussion, which the authors promised to include in their revision.

Overall, this paper seems to pass the bar for acceptance, and we strongly encourage the authors to imply the revisions discussed in their rebuttal in the camera ready version of this paper.